

# The bosonic skin effect: Boundary condensation in asymmetric transport

Louis Garbe[1*], Yuri Minoguchi[1], Julian Huber[1] and Peter Rabl[1,2,3,4]

**1** Vienna Center for Quantum Science and Technology, Atominstitut,
TU Wien, 1020 Vienna, Austria
**2** Technical University of Munich, TUM School of Natural Sciences,
Physics Department, 85748 Garching, Germany
**3** Walther-Meißner-Institut, Bayerische Akademie der Wissenschaften,
85748 Garching, Germany
**4** Munich Center for Quantum Science and Technology (MCQST), 80799 Munich, Germany

⋆ louis.garbe@wmi.badw.de

## Abstract

We study the incoherent transport of bosonic particles through a one dimensional lattice with different left and right hopping rates, as modelled by the asymmetric simple inclusion process (ASIP). Specifically, we show that as the current passing through this system increases, a transition occurs, which is signified by the appearance of a characteristic zigzag pattern in the stationary density profile near the boundary. In this highly unusual transport phase, the local particle distribution alternates on every site between a thermal distribution and a Bose-condensed state with broken $U(1)$-symmetry. Furthermore, we show that the onset of this phase is closely related to the so-called non-Hermitian skin effect and coincides with an exceptional point in the spectrum of density fluctuations. Therefore, this effect establishes a direct connection between quantum transport, non-equilibrium condensation phenomena and non-Hermitian topology, which can be probed in cold-atom experiments or in systems with long-lived photonic, polaritonic and plasmonic excitations.



# 1  Introduction

Transport phenomena are of relevance for almost all areas of physics and technology with transport of electric currents and heat conduction in solids being two prototypical examples. While electric currents are carried by electrons, i.e., massive fermionic particles, heat transfer can be understood as the emission and reabsorption of quantized lattice vibrations, i.e, non-conserved bosonic excitations. However, despite relying on very different microscopic mechanisms, both transport scenarios share many similarities. For example, depending on the mean free path, transport can either be ballistic or diffusive, where in the latter case Ohm's law and Fourier's law describe a similar linear relation between the current and the applied voltage or temperature gradient. Therefore, a general question of interest is under which conditions 'anomalous transport' with a qualitatively very different phenomenology can be observed.

In this paper, we consider the setup shown in Fig. 1 (a) as an elementary model to study dissipative transport of bosons. Here, bosons injected from a hot reservoir on the right can incoherently hop between neighboring sites of a one dimensional lattice, before being dumped into a second reservoir on the other end. This process has two key features: First, in the

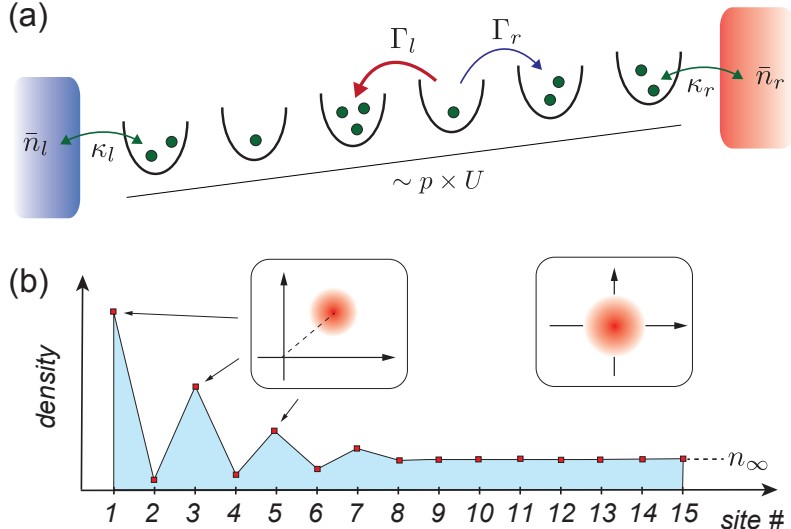

Figure 1: Asymmetric bosonic transport. (a) Sketch of the ASIP setup studied in this work. Bosons injected from a thermal particle reservoir with mean occupation number $\bar{n}_r$ on the right can incoherently hop along the lattice with asymmetric rates $\Gamma_l$ and $\Gamma_r$, before being emitted into a second reservoir with occupation number $\bar{n}_l$ on the left. A directional hopping can be imposed, for example, by applying a potential gradient with an energy offset $U$ between neighboring sites. (b) Under stationary conditions, this hopping asymmetry combined with the bosonic particle statistics results in the bosonic skin effect, i.e., the formation of a finite boundary region with a staggered density profile. The two insets show sketches of the Wigner distribution for individual lattice sites, indicating that within this boundary region, the odd sites are in a condensed state with broken $U(1)$ symmetry, while all other lattice sites exhibit a thermal distribution. See text for more details.

presence of a bias, the hopping rates to the left and right, $\Gamma_l$ and $\Gamma_r$, are in general different, in which case the transport is asymmetric, i.e., directional. Second, the hopping rates toward sites that are already occupied are enhanced by the bosonic particle statistics. Therefore, this process can be seen as the bosonic counterpart of the celebrated asymmetric simple exclusion process (ASEP) [1–4]—a common model for directed transport of fermions or classical hard-core particles—and one speaks of an asymmetric simple inclusion process (ASIP) instead.[1]

Compared to fermions as the carriers for electric currents, the dissipative transport of bosonic particles has attracted considerably less attention so far. This can be attributed to a lack of conventional solid-state systems where this physics could be observed. However, this situation has changed recently and a variety of experimental platforms have now become available where non-equilibrium processes with bosonic particles can be probed. This includes, for example, cold atoms in optical lattice potentials, where different techniques to study transport have already been demonstrated [9–13]. Furthermore, it has been shown in various experiments that long-lived photonic [14–16], polaritonic [17–23] or plasmonic [24] excitations can behave as massive bosonic particles and equilibrate with the surrounding material, before they eventually decay. The ongoing experimental advances in these platforms naturally raise the question of how transport in such settings is affected by the bosonic particle statistics of the carriers.

---

[1]Note that in the previous literature the term 'ASIP' is not uniquely defined and also used for a broader class of transport processes [5–8]. Here, ASIP exclusively refers to the direct bosonic analog of the ASEP, which describes incoherent hopping of actual bosonic particles.

In the following analysis, we investigate the properties of the ASIP in a thermal transport scenario, where we focus primarily on the stationary current and the density profile along the lattice. In the absence of asymmetry, we recover the usual diffusive transport in this model as well, characterized by a linear population gradient and a Fourier law for the current. However, as soon as a finite degree of asymmetry is introduced, the transport becomes ballistic and particles accumulate in a finite boundary region near the drain. Moreover, as the total current through the system increases, we observe a transition from a smooth pile-up to a zigzag structure, as depicted in Fig. 1 (b), with odd (even) sites being highly (weakly) populated. This phase represents a rather unusual non-equilibrium configuration, where the particle distribution alternates on every lattice site between a thermal distribution and that of a coherent state with broken $U(1)$-symmetry. This emergence of coherences in a purely dissipative and thermal transport scenario is very surprising and related to non-equilibrium condensation phenomena [14,19,25–28] that have no counterpart in fermionic transport. Therefore, we identify this boundary condensation as a unique feature of the ASIP model and call it the *bosonic skin effect*.

The observed accumulation of particles in a dissipative transport scenario is indeed very reminiscent of the so-called non-Hermitian skin effect (NHSE) [29–43]. This effect refers to the boundary localization of the eigenfunctions of certain non-Hermitian lattice Hamiltonians and is thus frequently discussed in connection with their topological classification [32, 35, 42, 44, 45]. However, such non-Hermitian models do not conserve the norm of the wavefunction nor the particle number. Therefore, beyond their mathematical interest, the relevance of the NHSE and other spectral features of non-Hermitian systems for actual quantum transport processes is not immediately clear and still a subject of ongoing investigations [31, 36–41, 43]. Here, by mapping the dynamics of density fluctuations in our system onto the paradigmatic Hatano-Nelson model (HNM) [29,35,42,45], we establish a direct correspondence between the eigenvalue structure of this non-Hermitian Hamiltonian and the stationary states of the ASIP transport problem. This correspondence relies on a subtle difference between Dirichlet and Neumann boundary conditions for the HNM and provides important additional insights into the nature of the predicted boundary transition. In particular, we find that the onset of condensation coincides with the appearance of a higher-order exceptional point in the HNM and occurs without a closing of the dissipative gap. This distinguishes the bosonic skin effect from other dissipative quantum phase transitions [46,47] and, in summary, reveals an unexpectedly rich interplay between transport, non-equilibrium condensation effects and non-Hermitian physics.

The remainder of the paper is structured as follows. In Sec. 2, we introduce the ASIP model and the main transport equations that we use to describe it. In Sec. 3, we present the bosonic skin effect and discuss the onset of the zigzag phase within mean-field theory, before investigating the full particle distribution and condensation effects in Sec. 4. Finally, in Sec. 5, we discuss the connection between the ASIP and the HNM, before summarizing our main findings in Sec. 6. Additional details about the analytic derivations and numerical methods are presented in the appendices.

## 2 Model

We consider the transport of bosons in a 1D lattice, as depicted in Fig. 1 (a). Here, the bosons are injected from a thermal reservoir on the right and propagate along a chain of $L$ lattice sites through incoherent hopping processes, before being emitted into a second reservoir on the left. In the following we are primarily interested in asymmetric transport, $\Gamma_l > \Gamma_r$, where $\Gamma_l$ and $\Gamma_r$ denote the hopping rates to the left and to the right, respectively.

## 2.1 The ASIP master equation

We model the dynamics of this system by the Lindblad master equation

$$\frac{d\hat{\rho}}{dt} = \left(\mathcal{L}_{\text{hop}} + \mathcal{L}_l + \mathcal{L}_r\right)\hat{\rho}, \tag{1}$$

where $\hat{\rho}$ is the system density operator. Here, the first term describes the incoherent hopping of bosons along the lattice. This process is described by the Liouville superoperator [40,48–50]

$$\mathcal{L}_{\text{hop}}\hat{\rho} = \sum_{p=1}^{L-1} \Gamma_l \mathcal{D}[\hat{a}_p^\dagger \hat{a}_{p+1}]\hat{\rho} + \Gamma_r \mathcal{D}[\hat{a}_{p+1}^\dagger \hat{a}_p]\hat{\rho}, \tag{2}$$

where $\hat{a}_p$ ($\hat{a}_p^\dagger$) are the bosonic annihilation (creation) operators for lattice site $p$ and we have introduced the short notation

$$\mathcal{D}[\hat{c}]\hat{\rho} = \hat{c}\hat{\rho}\hat{c}^\dagger - \frac{1}{2}\left(\hat{c}^\dagger \hat{c}\hat{\rho} + \hat{\rho}\hat{c}^\dagger \hat{c}\right). \tag{3}$$

In Eq. (2), the jump operator $\hat{a}_{p+1}^\dagger \hat{a}_p$ ($\hat{a}_{p-1}^\dagger \hat{a}_p$) destroys a boson at site $p$ and creates a boson at site $p+1$ ($p-1$) instead. This process conserves the total particle number and it is thus different from particle loss or gain. As a direct consequence of this particle number conservation, each jump operator is *quadratic* in $\hat{a}$ and $\hat{a}^\dagger$, and therefore the hopping process is nonlinear.

Note that in (2), the hopping dynamics is represented by an incoherent jump process, rather than the usual Hamiltonian evolution [29–43]. Below and in Appendix A, we discuss in more details how this incoherent dynamics can arise as an effective description, for instance, in systems of bosons hopping on a tilted lattice potential.

The second and the third term in Eq. (1) represent the coupling to the thermal particle reservoirs to the left and to the right, which we model by

$$\mathcal{L}_l\hat{\rho} = \kappa_l(\bar{n}_l + 1)\mathcal{D}[\hat{a}_1]\hat{\rho} + \kappa_l \bar{n}_l \mathcal{D}[\hat{a}_1^\dagger]\hat{\rho},$$
$$\mathcal{L}_r\hat{\rho} = \kappa_r(\bar{n}_r + 1)\mathcal{D}[\hat{a}_L]\hat{\rho} + \kappa_r \bar{n}_r \mathcal{D}[\hat{a}_L^\dagger]\hat{\rho}.$$

Here $\kappa_l$ and $\kappa_r$ denote the coupling rates to the two reservoirs and $\bar{n}_l$ and $\bar{n}_r$ are the corresponding thermal occupation numbers. Note that while we will only consider thermal baths in this work, other pumping mechanisms, such as incoherent gain, would result in a behavior that is qualitatively very similar to what is discussed below.

## 2.2 Asymmetric hopping

Before we proceed, let us briefly comment on the physical motivation behind this asymmetric transport model. A very generic scenario is depicted in Fig. 1 (a), where bosons are confined to a lattice with an energy gradient, for example, an optical lattice for cold atoms [10, 12], a nanophotonic lattice for exciton polaritons or plasmons [21, 24], etc. In this case, due to a large energy offset $U > 0$ between neighboring sites, coherent tunneling is suppressed, but in the presence of a phononic bath, the bosons may still transition between neighboring sites by emitting or absorbing vibrational excitations. Such a process can be modelled by a phonon-assisted tunneling term of the form

$$\hat{H}_{\text{int}} \sim \sum_p (\hat{a}_{p+1}^\dagger \hat{a}_p + \hat{a}_{p+1}\hat{a}_p^\dagger)(\hat{b}_p + \hat{b}_p^\dagger), \tag{4}$$

where the bosonic operators $\hat{b}_p$ represent local bath excitations. Roughly speaking, for a particle to jump to the left, it must lose the energy $\sim U$ by emitting it into the environment.

Conversely, to jump to the right, it must absorb the same amount of energy. Therefore, a bath at low temperature, where emission processes are more likely than absorption, favors hopping to the left.

More precisely, under the assumption that the bath is sufficiently Markovian, its dynamics can be eliminated to derive an equation of motion for the reduced system density operator $\hat{\rho}$ only. While some details may depend on the specific implementation (see Appendix A for a more detailed derivation), this master equation will be, quite generically, of the form given in Eq. (1), with hopping rates satisfying

$$\frac{\Gamma_l}{\Gamma_r} = \exp\left(\frac{\hbar U}{k_B T_{\text{phon}}}\right), \tag{5}$$

where $T_{\text{phon}}$ is the temperature of the phononic bath, which determines the asymmetry in this setting. Apart from such naturally occurring dissipative hopping mechanisms, there are also many systems where this asymmetric hopping processes can be engineered. For example, in optical lattices, directed dissipative hopping can be implemented via Raman processes [40,51–53], which involve atomic or cavity decay as a source of dissipation and directionality. Ideas for realizing number-conserving dissipation processes for photons have also been discussed for optomechanical systems [54,55] and circuit QED [56], and can be readily adapted for the implementation of directed hopping processes as well. In the following we do not consider any of these possible implementations specifically, but rather address the general properties of the transport model given in Eq. (1).

## 2.3 Transport

In this work we focus primarily on the stationary transport of particles between two thermal reservoirs. In the absence of asymmetry, transport would be solely driven by the temperature gradient between the reservoirs, i.e., by the difference between $\bar{n}_r$ and $\bar{n}_l$. For asymmetric rates, $\Gamma_l \neq \Gamma_r$, a directed particle flow develops even without any external temperature bias. To characterize transport in different parameter regimes, we consider the average stationary current $J$ as well as the stationary density profile $n_p = \langle \hat{n}_p \rangle = \langle \hat{a}_p^\dagger \hat{a}_p \rangle$ along the chain. Throughout this paper we adopt the convention that symbols with hats represent quantum operators, while symbols without hats denote their averages. Starting from the master equation in Eq. (1), the mean occupation number $n_p$ of any of the sites changes in time as

$$\frac{dn_p}{dt} = J_{p,p+1} - J_{p-1,p}. \tag{6}$$

This equation has the form of a conservation law, where, for any $p \in [1, N-1]$,

$$J_{p,p+1} = \Gamma_l \langle \hat{n}_{p+1}(1 + \hat{n}_p) \rangle - \Gamma_r \langle \hat{n}_p(1 + \hat{n}_{p+1}) \rangle, \tag{7}$$

is the average particle current between sites $p$ and $p+1$. Note that we have adopted the convention that a positive $J_{p,p+1}$ implies a current flowing from right to left, i.e., from site $p+1$ into site $p$. From Eq. (7) we already see that the current depends non-linearly on the density, due to bosonic bunching: indeed, the probability for a particle on site $p+1$ to jump to site $p$ is *enhanced* by a factor $1 + \hat{n}_p$, which depends on the population of the target site. On the boundaries, the currents

$$J_{0,1} = \kappa_l(n_1 - \bar{n}_l), \qquad J_{L,L+1} = \kappa_r(\bar{n}_r - n_L), \tag{8}$$

represent the flow of particles into the left bath and from the right bath, respectively. In the steady state, the particle current is conserved along the chain and we obtain

$$J_{p,p+1}(t \to \infty) = J, \qquad \forall \, p. \tag{9}$$

Note, however, that this uniformity of the current does not imply a uniform profile for the density $n_p$.

## 2.4 Mean-field dynamics

Although Eq. (1) contains only dissipative terms and no additional coherent interactions between the bosons, these incoherent processes are nonlinear and therefore do not permit a closed set of equations for the mean occupation numbers. In addition, since the number of possible bosonic configurations scales exponentially with the number of lattice sites $L$, brute-force numerical solutions of the master equation are also inaccessible for the parameter regimes of interest. Therefore, to proceed we resort to a mean-field decoupling of the equations of motions by factorizing expectation values as $\langle \hat{n}_p \hat{n}_{p+1} \rangle \approx \langle \hat{n}_p \rangle \langle \hat{n}_{p+1} \rangle$. Under this approximation, the average current reads

$$J_{p,p+1} \simeq \Gamma_l n_{p+1}(1+n_p) - \Gamma_r n_p(1+n_{p+1}). \tag{10}$$

The system is then described by a set of $L$ nonlinear differential equations, which can be solved efficiently numerically and also permit exact analytical solutions in the steady state.

To benchmark the validity of the mean-field approximation, we compare these predictions with exact Monte-Carlo simulations for small systems sizes and low occupation numbers $n_p \lesssim 1$ and with phase-space simulations based on the Truncated Wigner Approximation (TWA) [57] for larger occupation numbers. Within their respective regimes of validity, we find almost perfect agreement between the numerical results and the stationary distributions obtained from mean-field theory. Further details about these numerical methods and some of the benchmarks can be found in Appendix B.

## 2.5 Hydrodynamic limit

Additional insights about the transport dynamics in our system can be obtained by considering the continuum (or hydrodynamic) limit. To do so, we rewrite the mean-field equations of motion as

$$\frac{dn_p}{dt} = \frac{\Gamma_A}{2}(n_{p+1} - n_{p-1})(2n_p + 1) + \Gamma_S(n_{p-1} - 2n_p + n_{p+1}), \tag{11}$$

where $\Gamma_A = \Gamma_l - \Gamma_r$ and $\Gamma_S = (\Gamma_l + \Gamma_r)/2$. Then, under the assumption that the $n_p$ vary slowly between neighboring sites, we can replace them by a continuous field $n(x,t)$, where $x$ is the dimensionless position along the lattice. Away from the edges, this field obeys the partial differential equation

$$\partial_t n = \Gamma_A(1 + 2n)\partial_x n + \Gamma_S \partial_x^2 n. \tag{12}$$

This is, in essence, the well-known Burgers' equation [58–60], a simplified version of Navier-Stokes equation in hydrodynamics. The parameters $\Gamma_A$ and $\Gamma_S$ can thus be interpreted as non-linear advection and diffusion rates, respectively.

With the left side of the lattice being initially empty, the possible solutions of Eq. (12) include propagating shock fronts of the form [60]

$$n(x,t) = \frac{\bar{n}_{sw}}{2}\left[1 + \tanh\left(\frac{x - L + c_{sw}t}{w_{sw}}\right)\right], \tag{13}$$

where $\bar{n}_{sw}$ is the height, $c_{sw} = \Gamma_A(\bar{n}_{sw} + 1)$ the speed and $w_{sw} = 2\Gamma_S/(\bar{n}_{sw}\Gamma_A)$ the width of the wavefront. These solutions clearly illustrate how the bosonic enhancement factor affects transport. First, the velocity of the density wave scales with the typical density $\bar{n}_{sw}$. Second, the bosons in the high density region propagate faster than the bosons at the front, which leads to a compression of the wave and $w_{sw}$ going to 0 for very large $\bar{n}_{sw}$.

While the Burgers' equation provides valuable intuition about the transport dynamics in our system, it is based on a continuum approximation. As such, it is only expected to hold in a 'laminar' regime, i.e., when the effective Reynolds number

$$\text{Re} = \frac{\Gamma_A \bar{n}_{\text{sw}}}{\Gamma_S}, \tag{14}$$

associated with a typical occupation number $\bar{n}_{\text{sw}}$, is small.[2] In the opposite limit, the characteristic length scale, $w_{\text{sw}} \sim O(1)$, becomes of the order of the lattice spacing and new features can arise from the discreteness of the lattice and the presence of boundaries.

## 2.6 Relation to the ASEP

By replacing the bosonic operators in Eq. (1) by operators $\hat{a}_p$ that obey fermionic anti-commutation relations, i.e., $\{\hat{a}_p, \hat{a}_p^\dagger\} = 1$, we obtain the master equation describing the ASEP. In this case, the site occupation numbers $n_p$ obey the same equation as in Eq. (6), but with a fermionic current

$$J_{p,p+1}^{\text{ASEP}} = \Gamma_l \langle \hat{n}_{p+1}(1 - \hat{n}_p) \rangle - \Gamma_r \langle \hat{n}_p(1 - \hat{n}_{p+1}) \rangle. \tag{15}$$

Here, rather than being enhanced, the hopping to neighboring sites is prohibited by the Pauli exclusion principle, if the site is already occupied. The properties of the ASEP have been extensively studied in the literature [1–4]. This includes, most notably, the scaling of current fluctuations [61, 62] in infinite lattices, which falls into the Kardar-Parisi-Zhang (KPZ) universality class [4, 63, 64]. The ASEP is thus closely connected to surface growth and related non-equilibrium phenomena. It is therefore interesting to understand how the change from an exclusion to an inclusion process affects these properties. These aspects, however, will be discussed in more details elsewhere [65]. Instead, here we focus on novel effects that are unique to the ASIP and reveal themselves already at the mean-field level.

# 3 The bosonic skin effect

In the following section, we explore in more details the stationary state of the transport master equation in Eq. (1), which we describe in terms of the mean occupation numbers $n_p$ and the current $J$.

## 3.1 Transport regimes

In a first step, we show in Fig. 2 examples of the stationary density profile $n_p$ for a lattice of $L = 15$ sites, together with the scaling of the current $J$ as a function of $L$. From these plots we identify three qualitatively different transport regimes.

### 3.1.1 Diffusive transport

In Fig. 2 (a) we first consider the symmetric case $\Gamma_l = \Gamma_r$, where the stationary density profile along the chain is simply a linear interpolation between $\bar{n}_l$ and $\bar{n}_r$. This is also expected from

---

[2]In the literature, the Burgers' equation is often defined as $\partial_t n = n \partial_x n + \frac{1}{\text{Re}} \partial_x^2 n$, with Re the Reynolds number, and $n \sim O(1)$. In our case, Eq. (12) includes an additional linear term $\sim \Gamma_A \partial_x n$ and the field assumes values between 0 and $n_\infty$. However, we can rescale the field as $n \to n/n_\infty$ and time as $t \to 2\Gamma_A n_\infty t$ and remove the linear term by a Galilean transformation. After these transformations we recover the standard form of the Burgers' equation with Re given by Eq. (14).

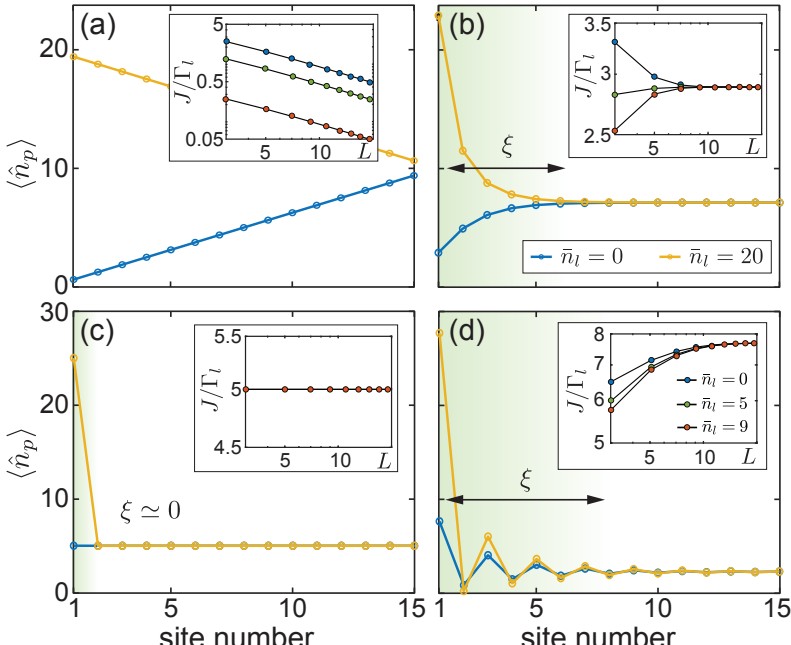

Figure 2: Plots of the steady-state occupation numbers $n_p$ for a lattice of $L = 15$ sites and different degrees of asymmetry: $\Gamma_A/\Gamma_l = 0$ (a), $\Gamma_A/\Gamma_l = 0.05$ (b), $\Gamma_A/\Gamma_l = 0.17$ (c), and $\Gamma_A/\Gamma_l = 1$ (d). For all plots $\bar{n}_r = 10$ and two different values of $\bar{n}_l = 0$ (blue lines) and $\bar{n}_l = 20$ (yellow lines) have been considered. The insets show the current $J$ versus the lattice size $L$, in log-log scale and for three different values of $\bar{n}_l = 0, 5, 9$. For $\Gamma_A = 0$ we recover a linear population gradient and the Fourier law for the current, as expected for diffusive transport. For any $\Gamma_A > 0$ and large $L$, the current becomes independent of both $L$ and $\bar{n}_l$, indicating ballistic transport. In this regime, we observe the formation of a finite boundary region of size $\xi$, as indicated by the shaded area. As the asymmetry increases, the width $\xi$ shrinks and vanishes for $\Gamma_A/\Gamma_l \simeq 0.17$. Beyond this point, a finite boundary region, but with an oscillating density profile, reappears. For all plots, we have set $\kappa_r = \kappa_l = \Gamma_l$.

Burgers' equation in the continuum limit, Eq. (12), which for symmetric hopping describes pure diffusion. In this regime, the current obeys the Fourier law and decreases with system size, i.e.,

$$J \propto \frac{\bar{n}_r - \bar{n}_l}{L}. \tag{16}$$

Interestingly, this diffusive transport is independent of the particle statistics and it is the same for bosons, fermions and noninteracting classical particles.

### 3.1.2 'Laminar' asymmetric transport

For a sufficiently large lattice, $L \gg 1$, the diffusive transport turns into directional transport for any finite hopping imbalance, $\Gamma_A \neq 0$. In this case, the stationary density profile is flat and assumes a constant value of $n_p \simeq n_\infty$ across most parts of the lattice. The exception is a region of size $\xi$ close to the left reservoir, where the density gradually adjusts to a boundary value, which depends on the occupation number of the left bath, $\bar{n}_l$. Most importantly, for a lattice size $L \gg \xi$, the stationary current $J > 0$ is completely independent of both $\bar{n}_l$ and the length of the chain [see the inset of Fig. 2 (b)]. This is true even though $\Gamma_r$ is still finite. This is in contrast to ballistic transport in coherent systems [66–69], where the stationary current

depends on the properties of both reservoirs.

While the quantitative details in this regime are already affected by the bosonically-enhanced hopping rates, the population profile is still qualitatively similar to what one would obtain for asymmetric hopping of independent classical particles. Moreover, since the effective Reynolds number introduced in Eq. (14) is still small, this behavior is well described by the continuous Burgers' equation in Eq. (12) and we can draw a close analogy with the regime of laminar flow in fluid dynamics.

### 3.1.3 'Turbulent' asymmetric transport

When either the asymmetry or the right bath occupation $\bar{n}_r$ are further increased, the size of the boundary region, $\xi$, decreases and reaches $\xi = 0$ at a critical value $\Gamma_A^c \equiv \Gamma_A^c(\bar{n}_r)$. At this specific point, the density profile is completely flat, with the exception of site $p = 1$, which is coupled to the left reservoir. As shown in the inset of Fig. 2 (c), since the relevant length scale vanishes, the current at this critical value is independent of the system size and adopts the value

$$J = \kappa_r \bar{n}_r \frac{\Gamma_l}{\Gamma_l + \kappa_r} \,. \tag{17}$$

Remarkably and somewhat unexpectedly, this situation occurs already for finite $\Gamma_r$, i.e., under conditions where particle flow in both directions is still possible.

As the directed particle flow is further increased, a boundary region of finite size $\xi$ reappears. In this regime, however, the occupation numbers vary strongly between neighboring sites and we observe a zigzag configuration with a decaying envelop. Counter-intuitively, as we keep increasing $\Gamma_A$, we find that the extent of this zigzag configuration *increases* in the direction *opposite* to the propagation. The transport in this regime is ballistic as well, i.e., for sufficiently large $L$ the current

$$J \approx \kappa_r \bar{n}_r > 0 \,, \tag{18}$$

is independent of both $\bar{n}_l$ and the system size. However, in contrast to the smooth pile-up observed above, this rapidly oscillating density profile is no longer captured by the Burgers' equation. This behavior is found for high effective Reynolds numbers and in analogy with turbulent flow in fluid dynamics, we observe a build-up of excitations at small length scales. In our discrete lattice setting, this leads to a breakdown of the continuum approximation.[3]

This staggered accumulation of particles in alternating lattice sites, rather than being distributed smoothly across the lattice, does not appear in analogous models for directed transport of fermions or classical particles. Since it arises from a purely dissipative process, this pattern must also be distinguished from the formation of standing waves in coherent channels [69]. It is thus a unique consequence of bosonic bunching.

## 3.2 Stationary density profile

Let us now proceed with a more in-depth analysis of the stationary density profile. In the steady state, the current $J$ is uniform across the lattice and we can use Eq. (10) to relate the occupation numbers between neighboring sites by

$$\Gamma_A n_p n_{p+1} + \Gamma_l n_{p+1} - \Gamma_r n_p = J \,, \tag{19}$$

for all $p$. For a large enough lattice, $L \gg 1$, and $p$ large, the occupation numbers near the right reservoir approach a constant value $n_p \sim n_{p+1} = n_\infty$, which is determined by the fixed point

---

[3]Note that similar oscillatory configurations are known from numerical simulations of the Burgers' equation, where they appear as pure discretization artefacts that must be avoided [59, 70]. In contrast, here we consider a discrete lattice to begin with and the formation of a staggered density profile is the main physical effect of interest.

of this equation. This leads to the following general relation,

$$J = \Gamma_A n_\infty (1 + n_\infty), \tag{20}$$

between the stationary current and the asymptotic particle density. The boundary condition for the reservoir on the right also gives us $J = \kappa_r(\bar{n}_r - n_\infty)$, which allows us to compute explicitly the asymptotic density,

$$n_\infty = \frac{1}{2}\sqrt{\left(1 + \frac{\kappa_r}{\Gamma_A}\right)^2 + \frac{4\bar{n}_r \kappa_r}{\Gamma_A}} - \frac{1}{2}\left(1 + \frac{\kappa_r}{\Gamma_A}\right), \tag{21}$$

and from it the stationary current $J$. Note that both quantities are smooth functions of all the system parameters and don't exhibit any sharp features. For large $\bar{n}_r$ we obtain $n_\infty \sim \sqrt{\bar{n}_r}$ and a current $J \approx \kappa_r \bar{n}_r$, which is limited by the influx of particles from the right reservoir.

The left boundary condition imposes $J = \kappa_r(n_1 - \bar{n}_r)$, meaning that $n_p \neq n_\infty$ for small site numbers $p$. In Appendix C we show in more details how the relation in Eq. (19) can be used to determine the full density profile $n_p$ in the limit $L \to \infty$, which for $\Gamma_l > \Gamma_r$ can be written in the form

$$\frac{n_p - n_\infty}{n_1 - n_\infty} = \left(\frac{\Gamma_S - c}{\Gamma_S + c}\right)^{p-1} \frac{1 + \left(\frac{\Gamma_S - c}{\Gamma_S + c}\right)\mu}{1 + \left(\frac{\Gamma_S - c}{\Gamma_S + c}\right)^p \mu}. \tag{22}$$

Here, $\mu$ is a constant that depends on the properties of the left reservoir, but its precise dependence is not important for the following discussion. In Eq. (22) we have also introduced the parameter

$$c = \Gamma_A\left(n_\infty + \frac{1}{2}\right), \tag{23}$$

which is the bosonically-enhanced speed of propagation. Indeed, $c$ is closely related to the speed of the shockwaves discussed in connection with the Burgers' equation (12), but determined by the self-adjusted, stationary density $n_\infty$.

By looking at the first term on the right side of Eq. (22), we see an exponential decay of the excess population, which can be re-expressed as

$$\left(\frac{\Gamma_S - c}{\Gamma_S + c}\right)^{p-1} = \begin{cases} e^{-\frac{p-1}{\xi}}, & \text{for} \quad c < \Gamma_S, \\ e^{-\left(\frac{1}{\xi} + i\pi\right)(p-1)}, & \text{for} \quad c > \Gamma_S. \end{cases} \tag{24}$$

Therefore, in both regimes, we can define the characteristic decay length

$$\xi = \frac{1}{\log\left|\frac{\Gamma_S + c}{\Gamma_S - c}\right|}. \tag{25}$$

As we increase $\Gamma_A$ or $\bar{n}_r$, $\xi$ decreases, and goes to zero for $c = \Gamma_S$. This allows us to identify the critical value of the hopping imbalance,

$$\frac{\Gamma_A^c}{\Gamma_l} = \frac{\Gamma_l + \kappa_r}{\Gamma_l + \kappa_r(1 + \bar{n}_r)}, \tag{26}$$

at which point $\xi = 0$ and the system changes between the smooth and the zigzag boundary configuration observed above. Beyond this point, we acquire an extra phase $\pi$, which explains the alternating occupation numbers for values of $\Gamma_A > \Gamma_A^c$. The full dependence of $\xi$ on $\Gamma_A$ and $\bar{n}_r$ is plotted in Fig. 3, which clearly shows a sharp drop to zero along the transition line $\Gamma_A = \Gamma_A^c$.

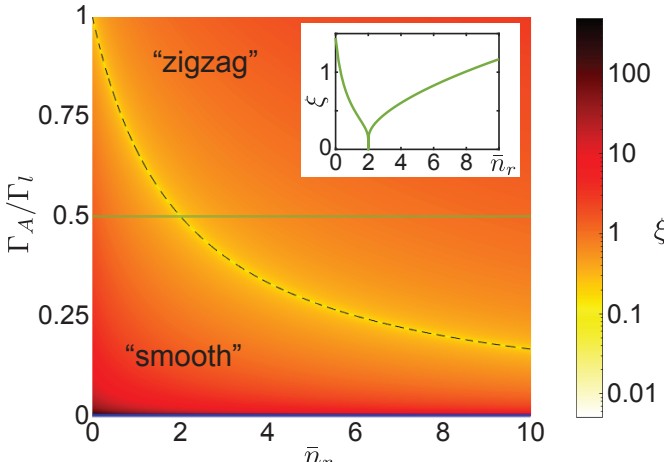

Figure 3: Dependence of the skin length $\xi$ as defined in Eq. (25) on the hopping asymmetry $\Gamma_A$ and on the thermal population of the right reservoir, $\bar{n}_r$. When $\Gamma_A$ is exactly zero (thick dark line at the bottom of the diagram), we recover the usual diffusive behavior. The dashed line corresponds to $\Gamma_A = \Gamma_A^c$, at which point $\xi = 0$. Below (above) this line, the steady-state population exhibits a smooth (zigzag) profile near the left boundary. The inset shows $\xi$ along the horizontal green line at $\Gamma_A = 0.5\Gamma_l$. For all points in this plot a value of $\kappa_r = \Gamma_l$ has been assumed, and the results are independent of both $\bar{n}_l$ and $\kappa_l$.

### 3.3 Nonlinear transport and the Fibonacci sequence

While the first term in Eq. (22) defines the characteristic size of the boundary region, it is important to keep in mind that the full density profile is not described by a simple exponential decay. This deviation, represented by the second term in Eq. (22), is due to the nonlinear nature of transport arising from the bosonic particle statistics. To obtain additional insights about this profile, we show in Appendix C that the stationary occupation numbers can be rewritten in the form

$$n_p = a\frac{y_{p-1}}{y_p} + d\,, \tag{27}$$

where the new quantities $y_p$ obey the recursion relation

$$y_{p+1} = ay_{p-1} + by_p\,, \tag{28}$$

with constants $a = (J\Gamma_A - \Gamma_l\Gamma_r)/\Gamma_A^2$, $b = 2\Gamma_S/\Gamma_A$ and $d = \Gamma_r/\Gamma_A$.

This reformulation shows that rather than being described by an exponential decay, the mathematical structure of $n_p$ is given by the ratio of successive coefficients of a generalized Fibonacci sequence defined by Eq. (28), also known as a Lucas sequence. For example, in the special case of $\Gamma_r = 0$ and a current $J = \Gamma_l$, we obtain $a = b = 1$ and $d = 0$ and the populations $n_p$ then oscillate toward $n_\infty = (1 + \sqrt{5})/\sqrt{2}$ in the same way that the ratio of successive coefficients of the Fibonacci sequence oscillates towards the golden ratio.

This observation is not just a purely mathematical curiosity, but a very generic feature of nonlinear transport. Indeed, *any* transport model with a next-neighbor nonlinear recursion relation of the type $\alpha n_p n_{p+1} + \beta n_p + \gamma n_{p+1} = \delta$ will lead to a density profile of the form given in Eq. (27). By contrast, recursion relations of the type $\beta n_p + \gamma n_{p+1} = \delta$, as encountered in linear transport models, give rise to a simple exponential population profile.

# 4 Boundary condensation

The strong bunching of the bosons in certain lattice sites, as observed for $\Gamma_A > \Gamma_A^c$, is somewhat similar to the formation of a Bose-Einstein condensate, where at low temperatures bosons tend to accumulate in a single momentum mode. However, in our setting this effect is observed under conditions where a large thermal current passes through the system, and locally one would expect a thermal distribution of particles instead. To resolve these two conflicting physical pictures, we must go beyond mean-field theory and take a closer look at the full particle number distributions and the coherence properties of our system.

## 4.1 Density fluctuations

To study effects beyond mean-field theory, we use numerical simulations based on the TWA. Within the TWA, the Wigner distribution is sampled by complex phase-space variables $\alpha_p$ that follow stochastic trajectories. Symmetrically-ordered expectation values of the form $\langle \hat{a}_p^{\dagger n} \hat{a}_q^m \rangle_{\mathrm{sym}}$ are then approximated by the corresponding stochastic averages $\langle \alpha_p^{*n} \alpha_q^m \rangle$. We refer to Appendix B for more details about this method. In Fig. 4 (a) we use the TWA to evaluate the equal-time two-particle correlation function

$$g_p^{(2)}(0) = \frac{\langle \hat{a}_p^\dagger \hat{a}_p^\dagger \hat{a}_p \hat{a}_p \rangle}{\langle \hat{a}_p^\dagger \hat{a}_p \rangle^2} \,, \tag{29}$$

for each of the lattice sites, and once the system has reached a steady state. The phase space plots below this curve show the corresponding distributions of the $\alpha_p$, as obtained from the individual trajectories in the numerical simulation. These sample the Wigner distribution of that site.

We see that, near the right reservoir, the value of this correlation function is $g^{(2)}(0) \simeq 2$, as expected for a thermal state [71]. The corresponding Wigner distributions are very close to a Gaussian distribution centered around $\alpha = 0$. Near the left boundary, however, $g^{(2)}(0)$ decreases for all odd sites and approaches a value of $g^{(2)}(0) \approx 1$, which indicates a coherent state. In this case the corresponding phase-space distribution has the shape of a symmetric ring with a maximum at a finite value of $|\alpha_p| \approx \sqrt{n_p}$. In contrast, on all even sites the distribution remains Gaussian-like and centered around $\alpha_p = 0$, although values of $g^{(2)}(0) > 2$ indicate small deviations from an exact thermal distribution. This overall behavior is further confirmed by the probability distributions $P(|\alpha_p|^2)$ plotted in Fig. 4 (b).

## 4.2 $U(1)$ symmetry breaking and phase coherence

The ASIP describes a purely incoherent hopping process. This means that the full master equation given in Eq. (1) is diagonal in the number basis and it is invariant under the local $U(1)$ symmetry transformations $\hat{a}_p \to \hat{a}_p e^{i\phi_p}$. This symmetry is also clearly visible in the phase-space plots in Fig. 4 (a), which are fully symmetric under rotation. However, these results only describe an ensemble average in the steady-state, while within a given experimental realization, or for finite time, the $U(1)$ symmetry can still be spontaneously broken.

To analyze potential symmetry-breaking effects in our system, we are interested in how long information about the phase in a given site is preserved. This is quantified by the coherence function

$$g_p^{(1)}(\tau) = \lim_{t \to \infty} \frac{\langle \hat{a}_p^\dagger(t + \tau) \hat{a}_p(t) \rangle_{\mathrm{sym}}}{\langle \hat{a}_p^\dagger(t) \hat{a}_p(t) \rangle_{\mathrm{sym}}} \,. \tag{30}$$

In Fig. 4 (c), we show the evolution of $g_p^{(1)}(\tau)$ as a function of the delay time $\tau$. We see that for odd sites near the left reservoir, this correlation function decays over a timescale $\tau_{\mathrm{coh}} \gtrsim 10\Gamma_l^{-1}$,

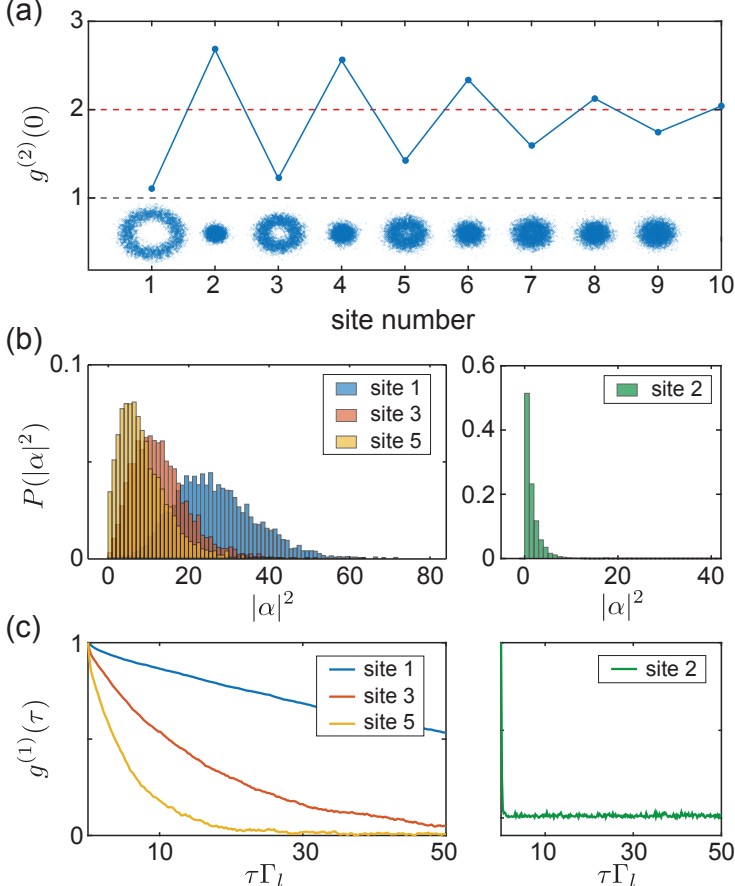

Figure 4: (a) Plot of the second-order correlation function $g^{(2)}(0)$ for a lattice of $L = 10$ sites, as obtained from a TWA simulation with 5000 trajectories. The phase-space distributions below each point indicate the distributions of the amplitudes $\alpha_p$, in the complex plane, at the final time of the simulation. (b) Distributions of the values of $|\alpha_p|^2$ and (c) plots of the coherence function $g^{(1)}(\tau)$ (30) for odd (left) and even (right) sites near the boundary (all the even sites have extremely similar behavior, here we show only $p = 2$ for simplicity). For all plots we have set $\kappa = \Gamma_l$, $\Gamma_r = 0$, $\bar{n}_r = 30$ and $\bar{n}_l = 0$. For the plots in (c), we have used a reference time of $t = 10\Gamma_l^{-1}$, which is sufficient to reach the steady state.

which is multiple times longer than the typical relaxation timescales in this system. In contrast, for even sites, no such extended phase correlation can be observed and the coherence vanishes on timescales much faster than $\Gamma_l^{-1}$. Note that we do not observe any significant cross-correlations between any of the lattices sites, either.

To understand this emergence of coherence in more details, we consider the totally asymmetric case, $\Gamma_r = 0$, and also assume $\bar{n}_l = 0$ for simplicity. Under these assumptions, the phase-space variable $\alpha_1$ of the first lattice site obeys the stochastic equation (see Appendix B.2)

$$d\alpha_1 = \frac{\Gamma_l n_2 - \kappa_l}{2} \alpha_1 dt + \sqrt{\frac{\kappa_l + \Gamma_l n_2}{2}} dW , \tag{31}$$

where $dW$ is a Wiener process. For the current discussion we have also adopted the convention $n_2 \equiv |\alpha_2|^2 - 1/2$ to be consistent with symmetrized expectation values, $\langle \hat{a}_p^\dagger \hat{a}_p \rangle_{\text{sym}} = n_p + 1/2 = \langle |\alpha_p|^2 \rangle$, even on the level of a single trajectory. Close to the steady state, the occupation number of the second site can be expressed in terms of the recursion

relation in Eq. (19) and approximated by

$$n_2(t) \simeq \frac{J/\Gamma_l}{1 + n_1(t)}.$$ (32)

After reinserting this results into Eq. (31), we obtain a closed diffusion equation for the variable $\alpha_1$, which is of the form

$$d\alpha_1 = \left( \frac{J}{|\alpha_1|^2 + 1/2} - \kappa_l \right) \frac{\alpha_1}{2} dt + \sqrt{D(\alpha_1)} dW.$$ (33)

From the deterministic part of this equation, we see that the nonlinear hopping process acts like an effective saturable gain. For sufficiently large $J$, this leads to a growth of the initial amplitude, which then saturates at a value $n_1 = |\alpha_1|^2 - 1/2 \sim J/\kappa_l$, consistent with the steady state result obtained from the mean-field analysis in this regime.

For $J/\kappa_l \gg 1$ and once the amplitude $\alpha_1$ has been amplified to a large value, it can be approximately written as $\alpha_1(t) \simeq \sqrt{n_1} e^{i\phi_1(t)}$, with a fixed $n_1$ and a phase $\phi_1(t)$ that obeys

$$d\phi_1 \simeq \sqrt{\frac{\kappa_l^2}{2J}} dW.$$ (34)

This phase diffusion equation predicts a decay of the ensemble-averaged amplitude according to

$$|\langle \alpha_1 \rangle| \propto e^{-t/\tau_{\mathrm{coh}}},$$ (35)

with a coherence time of $\tau_{\mathrm{coh}} = 4J/\kappa_l^2 \simeq 4\bar{n}_r/\kappa_l$. In Fig. 5 we consider a scenario in which the lattice is initialized in a symmetry-broken state, *i.e.*, a coherent state with a very small but finite amplitude on each site. The displacement direction can change from site to site, but remains the same from one trajectory to the next. For this initial configuration, the plots in Fig. 5 (a) show the successive evolution of the Wigner distribution of site $p = 1$. We clearly see that the small initial displacement is quickly amplified to its steady-state value, after which the phase diffuses on a much longer timescale. Eventually, we recover the ring-shaped profile shown in Fig. 4 (a). In Fig. 5 (b) we plot the evolution of $|\langle \alpha_1 \rangle|$ for different values of $\bar{n}_r$. The long-time decay of this quantity agrees very well with the analytic prediction in Eq. (35). Note also that, initially, each trajectory breaks the symmetry in the same direction, set by the initial perturbation. Hence, for intermediate times, the symmetry breaking is present even at the level of the density matrix.

## 4.3 Summary

In summary, the results presented in this section show that the zigzag structure observed at the mean-field level is consistent with the picture of an alternating lattice of condensed and thermal-like bosonic states. Consistently with other non-equilibrium condensation phenomena or closely related lasing effects [14,19,25–28], the Bose-condensed sites in our system are characterized by a spontaneously broken $U(1)$-symmetry with a phase coherence time that is long compared to the typical relaxation timescales in this system. The most surprising finding in our setting is that this effect occurs only in every other site near the boundary, while neighboring sites and other parts of the lattice remain close to a thermal state. This configuration is specific to the current transport scenario, where the stationary populations are determined by the nonlinear recursion relations discussed in Sec. 3.3, rather than by energetic considerations or an external gain mechanism.

Note that condensation effects have also been discussed for zero-range [5] and other attractive transport processes [5,6], where even on a periodic lattice all particles eventually

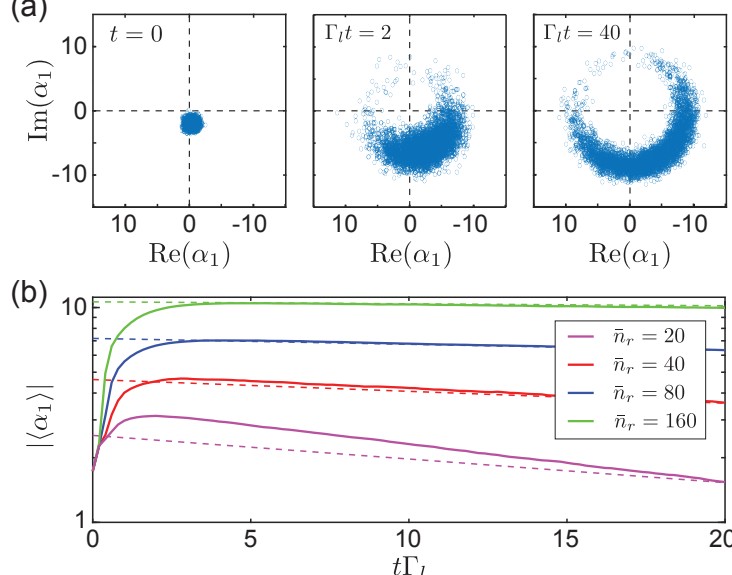

Figure 5: (a) Evolution of the Wigner distribution of site $p = 1$, when the system is initially prepared in a symmetry-broken state with $|\langle \alpha_p \rangle| = \sqrt{3}$ and a random phase. The three plots show the resulting phase-space distributions obtained in a TWA simulation for times $\Gamma_l t = 0, 2, 40$ and for $\bar{n}_r = 80$. On a short timescale, the initial displacement is amplified, while phase diffusion is observed over much longer times. (b) Logarithmic plot of the ensemble-averaged amplitude of the first site for the same initial conditions, but assuming different thermal occupation numbers of the right reservoir. After a short amplification, we observe an exponential decay of the average amplitude due to phase diffusion. The dashed lines represent the analytic prediction for this decay, as given in Eq. (35). For all plots, $\Gamma_l = \kappa_r$, $\Gamma_r = 0$, $L = 10$ and $\bar{n}_l = 0$.

accumulate in a single site. This is not the case for the ASIP considered here, where for periodic boundary conditions the system would simply evolve into an infinite-temperature state with all particle configurations being equally likely. Therefore, the presence of a boundary is essential to observe this type of condensation, which would not follow from an analysis of bulk properties only.

## 5 Asymmetric bosonic transport and the Hatano-Nelson model

As already pointed out in the introduction, the accumulation of particles near one end of the lattice in a dissipative transport model shares many similarities with the NHSE. This effect refers to the fact that the eigenfunctions of certain non-Hermitian lattice Hamiltonians, which are extended over the whole lattice for periodic boundary conditions, become exponentially localized when open boundary conditions are introduced. A prominent example where this effect occurs is the HNM [29], which, indeed, has originally been introduced to describe directional transport of bosons. However, in contrast to the full ASIP master equation considered here, the HNM is formulated in terms of a tight-binding Hamiltonian with asymmetric tunnelling amplitudes. Such a Hamiltonian is necessarily non-Hermitian, meaning that it does not preserve probabilities, particle numbers or operator commutation relations. Therefore, despite a considerable interest in the spectral properties of the HNM and related non-Hermitian Hamiltonians, their relevance for actual quantum transport problems often remains unclear.

While consistent embeddings of the HN Hamiltonian into a proper master equation have been discussed [31,37–39,41,43], these works have considered linear jump operators, which describe particles being exchanged with the environment. In this case, the evolution does not obey a conservation relation with a well-defined current, and the connection to the original dissipative hopping problem is lost. In the following we show instead how an explicit connection between the HNM and the ASIP transport problem can be established at the level of density fluctuations. This discussion complements the single-particle analysis of Ref. [40], and provides a new interpretation of the HNM at the level of a many-body transport problem. It also reveals a surprising relation between the dynamics of fluctuations and the stationary state of this system.

## 5.1   The non-Hermitian skin effect

The HNM is the simplest model to study boundary localization in non-Hermitian systems. It is described by the lattice Hamiltonian

$$\hat{H}_{\text{HN}} = i\sum_p J_l \hat{a}_p^\dagger \hat{a}_{p+1} - J_r \hat{a}_{p+1}^\dagger \hat{a}_p = \sum_{p,q} \hat{a}_p^\dagger (h_{\text{HN}})_{p,q} \hat{a}_q \,, \tag{36}$$

where the $\hat{a}_p$ represent non-interacting bosons or fermions, whose dynamics is then fully described by the tunneling matrix

$$h_{\text{HN}} = \begin{bmatrix} 0 & iJ_l & 0 & 0 & \dots & -ixJ_r \\ -iJ_r & 0 & iJ_l & 0 & 0 & \dots \\ 0 & -iJ_r & 0 & iJ_l & 0 & \dots \\ 0 & 0 & -iJ_r & 0 & iJ_l & \dots \\ \vdots & \vdots & \vdots & \ddots & \ddots & \ddots \end{bmatrix}. \tag{37}$$

Here, $x = 1$ for periodic boundary conditions and $x = 0$ for an open chain. For $J_r = J_l$ we recover the usual tight-binding Hamiltonian with real-valued single particle eigenenergies, $E_k = 2J_r \sin(k)$. The corresponding momentum eigenstates are extended over the whole lattice, both for open and periodic boundary conditions. For $J_r \neq J_l$, by contrast, the tunneling to the left and to the right is no longer the same, and $\hat{H}_{\text{HN}}^\dagger \neq \hat{H}_{\text{HN}}$. Still, when assuming periodic boundary conditions, the eigenfunctions of $h_{\text{HN}}$ remain plane waves, $\psi_k(p) \sim e^{-ikp}$, where $k \in [-\pi, \pi)$, but with a complex spectrum

$$E_k = (J_l + J_r)\sin(k) + i(J_l - J_r)\cos(k), \tag{38}$$

which describes an ellipse in the complex plane. In contrast, for open boundary conditions, all eigenmodes are exponentially localized near one end of the chain [72],

$$\psi_k(p) = (-i)^{p-1}\left(\frac{J_r}{J_l}\right)^{\frac{p-1}{2}} \sin(pk), \tag{39}$$

and are no longer orthogonal to each other. The corresponding spectrum is given by

$$E_k = 2\sqrt{J_r J_l}\cos(k). \tag{40}$$

Thus, the spectrum changes from a closed loop to a line in the complex plane (see Fig. 6 and the discussion below). This transition from an extended to a localized set of wavefunctions when changing from periodic to open boundary conditions occurs in many other related lattice models, and has been dubbed NHSE [29–43].

From Eq. (40) we see that when $J_r$ and $J_l$ have the same sign, i.e., $J_r J_l > 0$, the single-particle energies are real and therefore describe solutions that oscillate in time. However, when

$J_r J_l < 0$, the spectrum is purely imaginary, i.e., it describes decaying or amplified solutions. These two regimes are separated by a so-called exceptional point (EP) at $J_r = 0$, where the Hamiltonian of Eq. (36) becomes defective and cannot be diagonalized anymore. Instead, the tunneling matrix adopts a Jordan normal form

$$
h_{\text{HN}} = i J_l \begin{bmatrix} 0 & 1 & 0 & 0 & 0 & \dots \\ 0 & 0 & 1 & 0 & 0 & \dots \\ 0 & 0 & 0 & 1 & 0 & \dots \\ 0 & 0 & 0 & 0 & 1 & \dots \\ \vdots & \vdots & \vdots & \ddots & \ddots & \ddots \end{bmatrix},
\tag{41}
$$

which has only a single eigenmode with energy $E_{\text{EP}} = 0$ and a wavefunction $\psi_{\text{EP}}(p) = \delta_{p1}$, which is fully localized on the first site. The other basis elements are so-called *generalized eigenvectors*, i.e., they are transformed into $\psi_{\text{EP}}$ through the action of $h_{\text{HN}}$. The NHSE and the presence of exceptional points have recently attracted a lot of attention, in particular in connection with the classification of topological properties of non-Hermitian lattice systems [32, 35, 42, 44, 45].

## 5.2 Linearized boson transport

Let us now return to our mean-field model in Eq. (6) and consider a situation where at some initial time $t = 0$ the whole lattice is prepared in a state with a flat density distribution $n_p(0) = n_\infty$. For the successive evolution we make the Ansatz

$$
n_p(t) = n_\infty + \epsilon_p(t),
\tag{42}
$$

and assume that the fluctuations $\epsilon_p$ remain small compared to $n_\infty$. This is justified for short times and, more generally, under the condition $\bar{n}_r + \bar{n}_l \approx 2n_\infty$. We can then linearize the mean-field equations of motion and obtain

$$
\frac{d\epsilon_p}{dt} = c(\epsilon_{p+1} - \epsilon_{p-1}) + \Gamma_S(\epsilon_{p+1} + \epsilon_{p-1} - 2\epsilon_p)
$$
$$
+ \left[ (c + \Gamma_S - \kappa)\epsilon_p + \kappa \bar{m} \right] \delta_{p1} - (c - \Gamma_S + \kappa)\epsilon_p \delta_{pL},
\tag{43}
$$

with $\bar{m} = (\bar{n}_l + \bar{n}_r - 2n_\infty)$ and $\delta_{ij}$ the Kronecker delta, and we have set $\kappa_l = \kappa_r = \kappa$ for simplicity.

To connect this result to the HNM discussed above, we introduce the vectors $\vec{\epsilon} = (\epsilon_1, .., \epsilon_L)^T$ and $\vec{r}(\vec{\epsilon}) = (\bar{m} - \epsilon_1, 0, \dots, -\epsilon_L)^T$, such that

$$
\frac{d\vec{\epsilon}}{dt} = -ih\vec{\epsilon} + \kappa\vec{r},
\tag{44}
$$

with a non-Hermitian Hamiltonian

$$
h = i \begin{bmatrix} \Gamma_S + c & \Gamma_S + c & 0 & 0 & \dots \\ \Gamma_S - c & 0 & \Gamma_S + c & 0 & \dots \\ 0 & \Gamma_S - c & 0 & \Gamma_S + c & \dots \\ 0 & 0 & \Gamma_S - c & 0 & \dots \\ \vdots & \vdots & \vdots & \ddots & \ddots \end{bmatrix} - 2i\Gamma_S \mathbb{1}.
\tag{45}
$$

Therefore, ignoring the coupling to the reservoirs for now, i.e. $\kappa \to 0$, we see that the density fluctuations $\epsilon_p$ obey an effective Schrödinger equation with a non-Hermitian Hamiltonian $h$, which, by identifying $J_r \leftrightarrow c - \Gamma_S$ and $J_l \leftrightarrow c + \Gamma_S$, is very similar but not identical to $h_{\text{HN}}$. In particular, the diagonal elements of $h$ are shifted by a constant imaginary part $-2i\Gamma_S$ and, compared to $h_{\text{HN}}$, there is an additional term $c + \Gamma_S$ in the first entry of $h$. The first change merely shifts all the eigenenergies in the complex plane towards negative imaginary values, enforcing stable dynamics. The second change, as we will see, arises from the boundary conditions.

## 5.3 Neumann boundary conditions and the steady state

To understand the differences between $h$ and $h_{\mathrm{HN}}$, we emphasize that the dynamics of the fluctuations $\epsilon_p$ in Eq. (43) can still be written as a continuity equation,

$$\frac{d\epsilon_p}{dt} = j_{p,p+1} - j_{p-1,p}\,, \tag{46}$$

with currents

$$j_{p,p+1} = (\Gamma_S + c)\epsilon_{p+1} - (\Gamma_S - c)\epsilon_p\,. \tag{47}$$

This set of currents, $\vec{j} = (j_{1,2}, j_{2,3}, \dots)^T$, then obeys the equation of motion

$$\frac{d\vec{j}}{dt} = -i[h_{\mathrm{HN}} - 2i\Gamma_S \mathbb{1}]\vec{j}\,. \tag{48}$$

We see that, up to a global shift, it is the dynamics of current fluctuations that is governed by the non-Hermitian lattice Hamiltonian $h_{\mathrm{HN}}$ with *Dirichlet* boundary conditions

$$j_{0,1} = 0\,. \tag{49}$$

In other words, the linearized dynamics in our system is indeed governed by the HNM, but imposing *Neumann* boundary conditions for the density fluctuations $\epsilon_p$. This is physically consistent with the assumption $\kappa = 0$ made in this analysis.

This subtle change in the boundary conditions has an important consequence for the spectrum of $h$, namely the existence of a steady state. More precisely, in Appendix D we show that

$$\mathrm{Spec}\{h\}_L = \mathrm{Spec}\{h_{\mathrm{HN}} - 2i\Gamma_S\}_{L-1} \cup \{E_{\mathrm{ss}} = 0\}\,, \tag{50}$$

where $\mathrm{Spec}\{A\}_L$ is the spectrum of matrix $A$ in $L$ dimensions. This means, first of all, that the spectrum of density fluctuations in the ASIP model shares all the spectral features of the HNM, which we discussed in Sec. 5.1 above. In addition, there exists a unique steady state with $E_{\mathrm{ss}} = 0$ and a wavefunction

$$\psi_{\mathrm{ss}}(p) = \left(\frac{\Gamma_S - c}{\Gamma_S + c}\right)^{p-1}\,. \tag{51}$$

Up to nonlinear corrections, which have been omitted in the current analysis, this wavefunction agrees with the stationary density profile derived in Eq. (22). Note that the existence and the shape of this steady state does not change when the coupling to the reservoirs is no longer neglected, since the term $\sim \kappa(\epsilon_1 - \bar{m})$ in Eq. (43) merely fixes the magnitude of the fluctuation at the first site and $\epsilon_L \sim \psi_{\mathrm{ss}}(L) \to 0$.

## 5.4 Discussion

In Fig. 6, we plot the eigenvalues of $h_{\mathrm{HN}} - 2i\Gamma_S$, for Dirichlet and periodic boundary conditions, and compare them with the spectrum of $h$. These plots confirm that the eigenvalue structure of $h$ mimics that of the shifted HNM, except for the existence of a steady state with $E_{\mathrm{ss}} = 0$. For open lattices, the non-zero eigenvalues coalesce near $c = \Gamma_S$, which corresponds to the $(L-1)$-th order exceptional point EP for $J_r = 0$ in the HNM. By contrast, the steady-state mode remains well isolated and pinned at the origin. The explicit form of the steady-state solution in Eq. (51) confirms that this exceptional point coincides with the transition point into the zigzag phase in the full ASIP master equation. Hence, we have found a situation in which an EP for the *higher-energy* modes is directly connected with an observable configuration change in the *steady-state*.

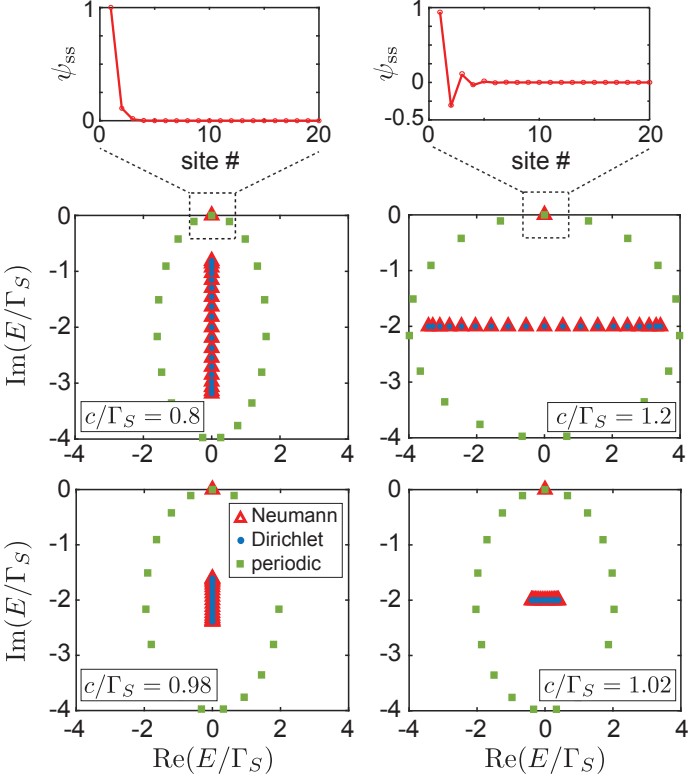

Figure 6: Complex spectrum of shifted HNM, $\tilde{h}_{\mathrm{HN}} = h_{\mathrm{HN}} - 2i\Gamma_S \mathbb{1}$, for different boundary conditions. The green squares ('periodic') and the blue dots ('Dirichlet') represent the eigenvalues of $\tilde{h}_{\mathrm{HN}}$ on a lattice of $L = 19$ sites with periodic and open boundary conditions, respectively. The red triangles ('Neumann') are the eigenvalues of $h$ as given in Eq. (45) for a lattice of $L = 20$ sites. The four lower panels show the complex spectra of these Hamiltonians for different values of $c$, increasing counter-clockwise. The spectrum of $h$ coincides with the one of $\tilde{h}_{\mathrm{HN}}$, plus the isolated steady-state at the origin. The two insets at the top depict the shape of the steady-state eigenmode $\psi_{\mathrm{ss}}$ before and after crossing the EP at a value of $c = \Gamma_S$. These results show that the transition into the zigzag structure of the steady state of $h$ coincides with the EP for its higher-energy modes.

In Ref. [40], this exponentially localized steady-state was also obtained, but only in the "smooth" phase, by considering the full spectrum of the hopping Liouvillian $\mathcal{L}_{\mathrm{hop}}$ restricted to the single-particle subspace. For a single boson, there are no nonlinearities, which corresponds to the limit $n_\infty \to 0$ and $c = \Gamma_A/2$. In this case, even in the fully asymmetric limit, $\Gamma_r \to 0$, the EP can be approached, but not crossed. A closely related analysis has also been performed for generalizations of the HNM with purely linear jump operators [39]. Here, the many-body stationary states for bosons and fermions exhibit an accumulation of particles near one boundary, but these features are rather broad and the direct connection to the NHSE has not been found there. We conclude that while many of these models show formally similar excitation spectra, the crossing of the EP and the transition into the zigzag phase is related to the nonlinearity of the underlying transport equations, which allows us to fulfill the condition $c > \Gamma_S$ through a bosonically enhanced propagation speed. At the same time, while being a many-body effect, for $\Gamma_r \to 0$ the zigzag pattern can already be observed deep in the quantum regime, i.e., for an average density of $n_\infty < 1$ (see Fig. 7).

The correspondence between the EP in the fluctuation dynamics and the transition in the stationary density profile is actually quite surprising. Naively one would expect that the EP, which occurs at an imaginary offset of $-2i\Gamma_S$, mainly influences the transient dynamics of decaying fluctuation modes. Instead, it signifies a sharp transition in the stationary fluctuation mode, which remains spectrally well isolated from the EP. This is in stark contrast to what is usually assumed for non-equilibrium phase transitions in dissipative systems, where the phase transition point coincides with a closure of the dissipative gap [46, 47]. The origin of this paradoxical situation can be traced back to the conservation of fluctuations, which, when decaying in site $p$, reappear in site $p - 1$. Fluctuations thus propagate across the chain, and fully decay only when they reach the edge. Therefore, while near the EP there is only a single eigenvalue that sets the timescale of the dynamics, it can still take a (diverging) time $\tau_{\text{relax}} \propto L$ for the system to fully relax. As pointed out in Refs. [36, 40, 43], it corresponds to the time required for the excitations to propagate along the chain. This distinguishes the analysis of such transport transitions from other non-equilibrium phase transitions in unbiased systems.

## 6 Summary and conclusions

In summary, we have studied the dissipative thermal transport of bosons through a lattice with asymmetric hopping rates, as described by the ASIP. Compared to analogous models for fermions or distinguishable particles, dissipative transport of bosons is characterized by hopping events that are accelerated by the presence of other particles. Our analysis showed that despite the simplicity of this process and without including any additional coherent interactions, this bosonic enhancement already gives rise to a highly non-standard transport phenomenology including ballistic currents, the formation of a boundary region with coexisting thermal and Bose-condensed sites, as well as the spontaneous development of coherence in a purely dissipative system.

In contrast to other condensation mechanisms that have been investigated for various (classical) inclusion processes [5, 6], the predicted transition for bosonic transport relies on the presence of a boundary and would be absent in an infinite or periodic lattice. This creates a natural connection to the HNM and related non-Hermitian lattice models, which we discussed in full details in Sec. 5. This analysis establishes a direct correspondence between the EP in the complex excitation spectrum of the HNM and the transition point in the stationary density profile of the ASIP. It also shows that, while closely related to the NHSE, the formation of the zigzag phase is a genuine many-body effect that does not appear for single-particle or linear transport models.

In conclusion, this bosonic skin effect creates an interesting connection between transport physics, non-equilibrium phase transitions and non-Hermitian physics. For highly asymmetric hopping rates, which can be engineered, for example, for cold atoms in optical lattices, the predicted zigzag phase is already observable at the level of a few atoms. Instead, in nanophotonic lattices for optical photons or exciton polaritons, where one might only achieve a small, temperature-induced bias, the necessary condition $c \sim \Gamma_S$ can still be reached by assuming pumped reservoirs with a considerably higher density. Therefore, since the main features associated with this transition are rather robust with respect to the details of the model, they should be observable in a variety of bosonic lattice systems, whenever hopping is predominantely incoherent.

## Acknowledgments

We thank A. Nunnenkamp, J.Schmitt, F. Roccati, F. Ciccarello, R. Filip and I. Egusquiza for stimulating discussions.

**Funding information** This work was supported by the Austrian Science Fund (FWF) through Grant No. P32299 (PHONED) and No. M3214 (ASYMM-LM), and the European Union's Horizon 2020 research and innovation programme under Grant Agreement No. 899354 (SuperQuLAN). This research is part of the Munich Quantum Valley, which is supported by the Bavarian state government with funds from the Hightech Agenda Bayern Plus.

## A Derivation of the transport master equation

In this section, we outline the derivation of the ASIP master equation in Eq. (1) for the case of a tilted lattice potential, where the bosons in each site are coupled to a bath of localized phonon modes. The Hamiltonian for this system can be written as

$$\hat{H} = -t_a \sum_{p=1}^{L-1} (\hat{a}_{p+1}^\dagger \hat{a}_p + \hat{a}_p^\dagger \hat{a}_{p+1}) + \sum_{p=1}^{L} pU\hat{a}_p^\dagger \hat{a}_p + \hat{H}_{\text{phon}}, \tag{A.1}$$

where $t_a$ is the tunneling amplitude and $U$ is the energy offset between two sites. The third term, $\hat{H}_{\text{phon}}$, accounts for the presence of the phononic bath, and we assume it to be of the form

$$\hat{H}_{\text{phon}} = \int_0^\infty d\omega \left[ \omega \hat{b}_{p,\omega}^\dagger \hat{b}_{p,\omega} + g(\omega)\hat{a}_p^\dagger \hat{a}_p (\hat{b}_{p,\omega} + \hat{b}_{p,\omega}^\dagger) \right]. \tag{A.2}$$

Here, the first part is the energy of the phononic modes with annihilation (creation) operators $\hat{b}_{p,\omega}$ ($\hat{b}_{p,\omega}^\dagger$) satisfying $[\hat{b}_{p,\omega}, \hat{b}_{q,\omega'}^\dagger] = \delta_{pq}\delta(\omega - \omega')$, and the second part describes a phonon-induced shift of each lattice site with some smooth coupling function $g(\omega)$.

In the limit $U \gg t_a$, coherent tunneling between neighboring sites is energetically suppressed and we can diagonalize the bare lattice Hamiltonian to lowest order in $\epsilon = t_a/U$. We do so by introducing the new bosonic operators

$$\hat{c}_p = \hat{a}_p + \epsilon(\hat{a}_{p+1} - \hat{a}_{p-1}) + O(\epsilon^2), \tag{A.3}$$

and write the full Hamiltonian as

$$\hat{H} \simeq \sum_p pU\hat{c}_p^\dagger \hat{c}_p + \int_0^\infty d\omega \, \omega \hat{b}_{p,\omega}^\dagger \hat{b}_{p,\omega} + \hat{H}_{\text{int}}. \tag{A.4}$$

To understand the effect of the remaining interaction term, $\hat{H}_{\text{int}}$, we move to the interaction picture and define

$$\hat{x}_p(t) = \int_0^\infty d\omega \, g(\omega) \left( \hat{b}_{p,\omega} e^{-i\omega t} + \hat{b}_{p,\omega}^\dagger e^{i\omega t} \right). \tag{A.5}$$

Then,

$$\hat{H}_{\text{int}}(t) = \sum_p \hat{x}_p(t) \left\{ \hat{c}_p^\dagger \hat{c}_p + \epsilon \hat{V}(t) + O(\epsilon^2) \right\}, \tag{A.6}$$

with

$$\hat{V}(t) = \left( \hat{c}_{p-1}^\dagger \hat{c}_p - \hat{c}_p^\dagger \hat{c}_{p+1} \right) e^{-iUt} + \text{H.c.} \tag{A.7}$$

We see that to zeroth-order in $\epsilon$, we only obtain an off-resonant energy shift $\hat{c}_p^\dagger \hat{c}_p$, which does not change the site occupation numbers and only leads to dephasing effects that depend on the bath spectral density at $\omega \approx 0$. To first order in $\epsilon$ we obtain a phonon-mediated hopping term, similar to Eq. (4).

After making a rotating wave approximation and keeping only the resonant terms in Eq. (A.6), we can eliminate the bath degrees of freedom and derive a master equation for the lattice bosons only. It is given by

$$\frac{d\hat{\rho}}{dt} \simeq \left( \mathcal{L}_{\text{deph}} + \mathcal{L}_{\text{hop}} \right) \hat{\rho} \,, \tag{A.8}$$

where

$$\mathcal{L}_{\text{deph}} = \sum_p \Gamma_\Phi \mathcal{D}[\hat{n}_p] \,, \tag{A.9}$$

is a pure dephasing term and

$$\mathcal{L}_{\text{hop}} = \Gamma_l \mathcal{D}[\hat{c}_{p-1}^\dagger \hat{c}_p - \hat{c}_p^\dagger \hat{c}_{p+1}] + \Gamma_r \mathcal{D}[\hat{c}_{p-1}^\dagger \hat{c}_p - \hat{c}_p^\dagger \hat{c}_{p+1}] \,, \tag{A.10}$$

accounts for the incoherent, phonon-mediated hopping between neighboring sites. In these expressions, $\Gamma_\Phi = C_{xx}(0)$ and

$$\Gamma_l = \epsilon^2 C_{xx}(U) \,, \qquad \Gamma_r = \epsilon^2 C_{xx}(-U) \,, \tag{A.11}$$

where

$$C_{xx}(\omega) = \int_0^\infty ds\, e^{i\omega s} \langle \hat{x}(t)\hat{x}(t-s) \rangle \,, \tag{A.12}$$

is the correlation spectrum of the phonon bath. When the bath is in a thermal state with temperature $T_{\text{phon}}$, we obtain

$$\frac{C_{xx}(\omega)}{C_{xx}(-\omega)} = e^{\hbar\omega/(k_B T_{\text{phon}})} \,, \tag{A.13}$$

which leads to the relation between the hopping rates given in Eq. (5). Note that in Eq. (A.8) we have omitted additional cross-site dephasing terms, which scale as $\sim \epsilon^2 \Gamma_\Phi$ and can therefore be neglected compared to $\mathcal{L}_{\text{deph}}$.

Due to the simple structure of the bath considered in this model, Eq. (A.10) still contains cross-terms of the form $(\hat{c}_{p-1}\hat{c}_p^\dagger)\hat{\rho}(\hat{c}_{p+1}\hat{c}_p^\dagger)$, which involve the coherences of the density matrix. These coherences, however, will be washed out under the influence of $\mathcal{L}_{\text{deph}}$ or any additional dephasing terms that might appear in a more realistic setting. We emphasize that the presence of such dephasing terms has no influence on the population dynamics, the particle currents or the stationary density profiles investigated in this work. Therefore, we conclude that the master equation given in Eq. (1) is indeed a rather generic model to study dissipative bosonic transport. Note that this does not apply to the coherence functions evaluated in Sec. 4, which are sensitive to $\Gamma_\Phi$ and thus to specific details of the environment.

# B  Numerical methods

## B.1  Low density regime: Monte-Carlo simulations

In the absence of any additional Hamiltonian terms, the master equation in Eq. (1) is diagonal in the Fock basis $|\{\vec{n}\}\rangle = |n_1, n_2 .., n_L\rangle$. The configuration is encoded by the vector $\vec{n} = (n_1, .., n_L)^T$, where the $n_p$ denote the number of bosons in each site. Therefore, we can restrict our analysis to the diagonal elements of the density operator, $P(\{\vec{n}\}, t) = \langle \{\vec{n}\}| \hat{\rho}(t) |\{\vec{n}\}\rangle$,

which describe the probabilities of different particle configurations. These probabilities evolve as

$$
\begin{aligned}
\dot{P} =& \Gamma_l \sum_p n_p(1+n_{p+1})P(\{\vec{n}+\vec{\delta}_{p,p+1}\}) - (1+n_p)n_{p+1}P \\
& + \Gamma_r \sum_p n_{p+1}(1+n_p)P(\{\vec{n}-\vec{\delta}_{p,p+1}\}) - n_p(1+n_{p+1})P \\
& + \kappa_l \Big\{ \bar{n}_l n_1 P(\{\vec{n}-\vec{\epsilon}_1\}) + (\bar{n}_l+1)(n_1+1)P(\{\vec{n}+\vec{\epsilon}_1\}) \\
& \qquad - [\bar{n}_l(1+n_1)+(\bar{n}_l+1)n_1]P \Big\} + (l \leftrightarrow r),
\end{aligned}
$$

where the last term is obtained by doing the substitution $(l \leftrightarrow r)$, $\vec{\epsilon}_1 \leftrightarrow \vec{\epsilon}_L$, and $n_1 \leftrightarrow n_L$. Here, $\epsilon_p^j = \delta_{pj}$, $\vec{\delta}_{p,p+1} = \vec{\epsilon}_{p+1} - \vec{\epsilon}_p$, we used a short notation $P = P(\{\vec{n}\})$, and omitted time dependence to lighten the notations.

Due to the exponentially growing configuration space, the exact dynamics of $P(\{n_p\}, t)$ can only be calculated for very small lattices and low occupation numbers. Instead, for larger lattices we sample the probability distribution via a Monte-Carlo simulation. To do so, the boson numbers $n_p(t)$ for each site are treated as stochastic variables, which during an infinitesimal time step $dt$ evolve according to

$$
dn_p = dN_p^l - dN_{p-1}^l + dN_{p-1}^r - dN_p^r. \tag{B.1}
$$

Here, the $dN_p^{l,r} = 0, 1$ are independent random variables and indicate that a boson has hopped to the left (right) when $dN_p^l = 1$ ($dN_p^r = 1$). The probabilities for these events are

$$
p(dN_p^l = 1) = \Gamma_l n_{p+1}(1+n_p)dt, \tag{B.2}
$$

$$
p(dN_p^r = 1) = \Gamma_r(1+n_{p+1})n_p dt, \tag{B.3}
$$

and $p(dN_p^i = 0) = 1 - p(dN_p^i = 1)$. By starting from a given initial configuration, $\{n_p(t=0)\}$, and evolving a total number of $\mathcal{N}_t$ stochastic trajectories in time, we can approximate the expectation value of any function of operators $\hat{n}_p$ by an ensemble average. For example,

$$
\langle \hat{n}_p \hat{n}_q \rangle(t) \simeq \frac{1}{\mathcal{N}_t} \sum_{i=1}^{\mathcal{N}_t} n_p(t)n_q(t) =: \langle n_p(t)n_q(t) \rangle. \tag{B.4}
$$

This method becomes exact in the limit $\mathcal{N}_t \to \infty$, and therefore also accounts for cross-site correlations, $C_{pq}(t) = \langle \hat{n}_p \hat{n}_q \rangle(t) - \langle \hat{n}_p \rangle(t)\langle \hat{n}_q \rangle(t)$, which are neglected in mean-field theory. It cannot, however, be used to predict quantities such as cross-site coherences of the form $\langle \hat{a}_p^\dagger \hat{a}_{p+1} \rangle$, because those involve off-diagonal elements of the density matrix. Furthermore, this method is limited to low average occupation numbers, since otherwise the rate of jumps, and therefore also the total simulation time, increases significantly.

## B.2  High density regime: Truncated Wigner approximation

The TWA is a technique for simulating the dynamics of bosons in phase space, which is spanned by complex amplitudes $\alpha_p$ and $\alpha_p^*$ defined on each site $p$. The state of the full lattice is then fully described by a multi-mode Wigner distribution $W(\{\alpha_p\}, t)$ on this space and expectation values of symmetrically-ordered operator products can be obtained from the moments of this function. For example,

$$
\langle \hat{a}_p^{\dagger n} \hat{a}_q^m \rangle_{\text{sym}} = \int d^{2L}\alpha \, (\alpha_p^*)^n \alpha_q^m W(\{\alpha_p\}). \tag{B.5}
$$

To obtain the equation of motion for $W(\{\alpha_p\})$ we use the substitutions [57]

$$\hat{a}_p^\dagger \hat{\rho} \rightarrow \left( \alpha_p^* - \frac{1}{2}\partial_{\alpha_p} \right) W, \qquad \hat{a}_p \hat{\rho} \rightarrow \left( \alpha_p + \frac{1}{2}\partial_{\alpha_p^*} \right) W,$$

etc., to convert the master equation (1) for the density operator into a partial differential equation for $W$. To illustrate this approach, let us consider only a single term, $\frac{d\hat{\rho}}{dt} = \Gamma_l \mathcal{D}[\hat{a}_1^\dagger \hat{a}_2]\hat{\rho}$, which translates into

$$\begin{aligned}
\frac{\partial W}{\partial t} = \frac{\Gamma_l}{2} \Big\{ & \partial_1 \left( \frac{\alpha_1}{2} - \alpha_1 |\alpha_2|^2 \right) + \partial_2 \left( \frac{\alpha_2}{2} + \alpha_2 |\alpha_1|^2 \right) \\
& + \partial_1^* \partial_1 \left( \frac{|\alpha_2|^2}{2} - \frac{1}{4} \right) + \partial_2 \partial_2^* \left( \frac{|\alpha_1|^2}{2} + \frac{1}{4} \right) \\
& - \partial_1 \partial_2 \alpha_1 \alpha_2 + \frac{1}{4}\partial_1 \partial_1^* \partial_2 \alpha_2 - \frac{1}{4}\partial_2 \partial_2^* \partial_1 \alpha_1 + c.c. \Big\} W,
\end{aligned} \tag{B.6}$$

where we have used the short-hand notation $\partial_i = \partial_{\alpha_i}$, and $\partial_i^* = \partial_{\alpha_i^*}$. Note that the same equation was derived in [73], where the two bosonic modes represented Schwinger bosons describing a $d$-level system.

The TWA consists in neglecting in this equation all third-order derivatives. This approximation is expected to be accurate when the number of bosons in the chain is high (see [57,74] for a more detailed discussion). Hence, this method provides a complementary treatment to the one presented in the previous section. After we performed the TWA, we obtain a Fokker-Planck equation, governed by a drift vector $\vec{A}$ and a diffusion matrix $D$,

$$\frac{\partial W}{\partial t} = -\partial_\lambda (A_\lambda W) + \frac{1}{2}\partial_\lambda \partial_\mu^* (D_{\lambda\mu} W), \tag{B.7}$$

where we have used Einstein's sum convention and the $2L$ greek indices run over all $\alpha_p$ and $\alpha_p^*$. For the example given in Eq. (B.6) above, the corresponding diffusion matrix is given by

$$D = \frac{\Gamma_l}{2} \begin{pmatrix} |\alpha_2|^2 & -\alpha_1 \alpha_2 & 0 & 0 \\ -\alpha_1^* \alpha_2^* & |\alpha_1|^2 & 0 & 0 \\ 0 & 0 & |\alpha_2|^2 & -\alpha_1^* \alpha_2^* \\ 0 & 0 & -\alpha_1 \alpha_2 & |\alpha_1|^2 \end{pmatrix},$$

where we have ordered the four independent variables as $(\alpha_1, \alpha_2^*, \alpha_1^*, \alpha_2)$. We have also omitted the constant terms $\pm 1/4$, which cancel when adding the contributions from all lattice sites, expect at the boundaries. For any other site, the diffusion matrix is positive semi-definite and can therefore be written as $D = BB^\dagger$ with

$$B = \sqrt{\frac{\Gamma_l}{4}} \begin{pmatrix} \alpha_2 & -\alpha_2 & 0 & 0 \\ -\alpha_1^* & \alpha_1^* & 0 & 0 \\ 0 & 0 & \alpha_2^* & -\alpha_2^* \\ 0 & 0 & -\alpha_1 & \alpha_1 \end{pmatrix}.$$

Therefore, it is possible to unravel the Fokker-Planck equation in terms of stochastic trajectories in phase space, which follow the (Ito) equations

$$d\vec{\alpha}_\lambda = \vec{A}_\lambda dt + B_{\lambda\mu} d\vec{W}_\mu. \tag{B.8}$$

Here, $\vec{W} = (W_1, W_2^*, W_1^*, W_2)$, where the $dW_i$ are complex-valued Wiener processes satisfying $\langle dW_i dW_i^* \rangle = 1$ and $\langle dW_i dW_i \rangle = \langle dW_i \rangle = 0$. By defining $dV = (dW_1 - dW_2^*)/\sqrt{2}$, we can write

the stochastic equations as

$$d\alpha_1 = \frac{\Gamma_l}{2}\alpha_1\left(|\alpha_2|^2 - \frac{1}{2}\right)dt + \sqrt{\frac{\Gamma_l}{2}}\alpha_2 dV,$$

$$d\alpha_2 = -\frac{\Gamma_l}{2}\alpha_2\left(|\alpha_1|^2 + \frac{1}{2}\right)dt - \sqrt{\frac{\Gamma_l}{2}}\alpha_1 dV^*.$$

This derivation can be generalized in a straightforward manner to all lattice sites and including the hopping to the right and the coupling to the reservoirs. Altogether we end up with the following set of stochastic differential equations

$$d\alpha_p = \frac{\Gamma_A}{2}\alpha_p\left(|\alpha_{p+1}|^2 - |\alpha_{p-1}|^2\right)dt - \Gamma_S\alpha_p dt + \sqrt{\Gamma_S}\left(\alpha_{p+1}dV_p - \alpha_{p-1}dV_{p-1}^*\right), \tag{B.9}$$

$$d\alpha_1 = \frac{\Gamma_A}{2}\alpha_1|\alpha_2|^2 dt - \frac{\Gamma_S}{2}\alpha_1 dt + \sqrt{\Gamma_S}\alpha_2 dV_1 - \frac{\kappa_l}{2}\alpha_1 dt + \sqrt{\frac{\kappa_l}{2}(2\bar{n}_l + 1) - \frac{\Gamma_A}{4}}dV_l,$$

$$d\alpha_L = -\frac{\Gamma_A}{2}\alpha_L|\alpha_{L-1}|^2 dt - \frac{\Gamma_S}{2}\alpha_L dt - \sqrt{\Gamma_S}\alpha_{L-1}dV_{L-1}^* - \frac{\kappa_r}{2}\alpha_L dt + \sqrt{\frac{\kappa_r}{2}(2\bar{n}_r + 1) + \frac{\Gamma_A}{4}}dV_r,$$

where all the $dV_i$ are independent complex Wiener processes. Note that in the equation for $\alpha_1$, the diffusion rate in the last term can become negative, when the coupling to the left reservoirs is too weak. This problem does not occur in any of the presented results, where we assume $\kappa_l = \Gamma_l$. In this case, the noise processes $\sim dV_1$ and $\sim dV_l$ can be combined in a single stochastic process, and for $\Gamma_r = \bar{n}_l = 0$ we obtain Eq. (31).

### B.3 Benchmarking the mean-field approximation

In Fig. 7, we compare the results obtained with these two numerical methods with the predictions from mean-field theory in the limits of low and high occupation numbers. These plots show that all the features in the stationary density profile discussed in the main text are accurately reproduced by both methods, within their respective range of applicability. In particular, the exact results from the Monte-Carlo simulations demonstrate that the predicted density patterns are already visible in parameter regimes where there is on average less than one boson per site. We also find that the mean-field prediction for the transition point $\Gamma_A^c$ is well reproduced by both methods (not shown here).

## C Derivation of the stationary density profile

In this section we provide additional details about the derivation of the steady-state occupation numbers $n_p$ within the mean-field approximation. The starting point for this derivation is Eq. (19), which for $L \to \infty$ already determines the relation between the current $J$ and the asymptotic occupation number $n_\infty$, as given in Eq. (20). To solve the full recursion relation, we first introduce a new variable $v_p = n_p - \Gamma_r/\Gamma_A$, which obeys

$$v_{p+1} = \frac{a}{v_p + b}, \tag{C.1}$$

with $a = (J\Gamma_A - \Gamma_l\Gamma_r)/\Gamma_A^2$ and $b = 2\Gamma_S/\Gamma_A$. In a next step, we make the Ansatz $v_p = ay_{p-1}/y_p$ to obtain a new sequence of numbers $y_p$, which satisfy

$$y_{p+1} = ay_{p-1} + by_p. \tag{C.2}$$

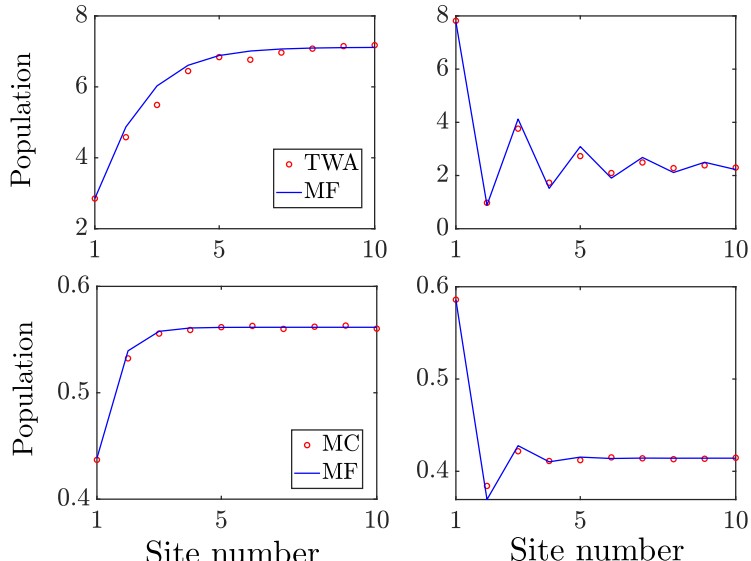

Figure 7: Top row: Comparison between the steady-state populations as predicted by mean-field (MF) theory and by the TWA, for $\bar{n}_r = 10$ and for $\Gamma_r/\Gamma_l = 0.95$ (left) and $\Gamma_r = 0$ (right). Bottom row: Comparison between the steady-state populations as predicted by mean-field theory and by exact Monte-Carlo (MC) simulations, for $\bar{n}_r = 1$ and $\Gamma_r/\Gamma_l = 0.5$ (left) and $\Gamma_r = 0$ (right). For these results we simulated $5 \times 10^3$ trajectories for the TWA and $5 \times 10^5$ trajectories for the Monte-Carlo method. For all plots, $\kappa_r = \kappa_l = \Gamma_l$, $\bar{n}_l = 0$, and $L = 10$.

Hence, the $y_p$ are given by a generalization of the Fibonacci sequence (known as the Lucas sequence). We can express the elements of this sequence as

$$y_p = \alpha \phi_+^p + \beta \phi_-^p, \tag{C.3}$$

where the constants $\alpha$ and $\beta$ depend on the initial condition and

$$\phi_\pm = \frac{b \pm \sqrt{b^2 + 4a}}{2} = \frac{1}{\Gamma_A}(\Gamma_S \pm c), \tag{C.4}$$

with $c = \Gamma_A(1 + 2n_\infty)/2$. To obtain this last equality, we used the boundary condition $J = \Gamma_A n_\infty(1 + n_\infty)$, from which it follows that $\sqrt{4a + b^2} = 2n_\infty + 1$. Therefore, in terms of these quantities, we obtain a general expression for the mean occupation number of each site,

$$
\begin{aligned}
n_p &= a \frac{\alpha \phi_+^{p-1} + \beta \phi_-^{p-1}}{\alpha \phi_+^p + \beta \phi_-^p} + \frac{\Gamma_r}{\Gamma_A} \\
&= \left(n_\infty - \frac{\Gamma_r}{\Gamma_A}\right) \frac{1 + \left(\frac{\Gamma_S - c}{\Gamma_S + c}\right)^{p-1} \mu}{1 + \left(\frac{\Gamma_S - c}{\Gamma_S + c}\right)^p \mu} + \frac{\Gamma_r}{\Gamma_A},
\end{aligned}
\tag{C.5}
$$

where $\mu = \beta/\alpha$. By rewriting the above result in terms of the ratio $(n_p - n_\infty)/(n_1 - n_\infty)$ we obtain Eq. (22), from which the decay of the zigzag structure becomes more obvious.

At this point, the parameters $n_\infty$ and $\mu$ are still unknown and must be determined by the boundary conditions. Since in the steady-state the current is constant we obtain

$$J = \kappa_l(n_1 - \bar{n}_l) = \kappa_r(\bar{n}_r - n_L) = \Gamma_A n_\infty(1 + n_\infty).$$

In the limit of a large lattice, we can set $n_L = n_\infty$, which gives us a quadratic equation for $n_\infty$,

$$\Gamma_A n_\infty^2 + (\Gamma_A + \kappa_r)n_\infty = \kappa_r \bar{n}_r \,,$$

with a solution displayed in Eq. (21). Finally, from the current into the left reservoir and the result for $n_{p=1}$ in Eq. (C.5) we can determine the value of $\mu$. For $\bar{n}_l = 0$ and $\Gamma_r = 0$, its explicit expression is

$$\mu = \left(\frac{\Gamma_S + c}{\Gamma_S - c}\right)\frac{2n_\infty - \bar{n}_r}{\bar{n}_r - n_\infty} \,.$$

In the most general case, its precise functional dependence on all the system parameters is complicated and of limited interest.

## D  Eigenstates of HNM with Neumann boundary conditions

In this appendix we derive the relation between the spectra of $h_{\mathrm{HN}}$ and $h$, which correspond to the HNM with Dirichlet and Neumann boundary conditions, respectively. An alternative derivation, and further results on these kinds of matrices, can be found in [72]. We introduce here the $L$-dimensional current vector $\vec{j} = (j_{0,1}, j_{1,2}, j_{2,3}, ...)^T$, which includes the component $j_{0,1} = 0$. Then, according to Eq. (47), we obtain the linear relation

$$\vec{j} = V\vec{\epsilon}\,, \tag{D.1}$$

between current and density fluctuations, where

$$V = \begin{bmatrix} 0 & 0 & 0 & 0 & \dots \\ c - \Gamma_S & \Gamma_S + c & 0 & 0 & \dots \\ 0 & c - \Gamma_S & \Gamma_S + c & 0 & \dots \\ 0 & 0 & c - \Gamma_S & \Gamma_S + c & \ddots \\ \vdots & \vdots & \vdots & \ddots & \ddots \end{bmatrix}\,. \tag{D.2}$$

By comparing Eq. (D.1) with the continuity equation (46), we find that

$$h = i\nabla V\,, \tag{D.3}$$

where $\nabla$ with $\nabla_{ij} = \delta_{ij-1} - \delta_{ij}$ is the discrete gradient. It follows that

$$\frac{d\vec{j}}{dt} = -i(iV\nabla)\vec{j}\,, \tag{D.4}$$

where the effective Hamiltonian for the current is of the form

$$iV\nabla = \left(\begin{array}{c|ccc} 0 & 0 & 0 & \dots \\ \hline c - \Gamma_S & & & \\ 0 & & h_{\mathrm{HN}} - 2i\Gamma_S & \\ \vdots & & & \end{array}\right)\,. \tag{D.5}$$

This is the result given in Eq. (48), but with the $j_{0,1}$ component included. Let us now consider a vector $\vec{\Phi}_k = (0, \vec{\psi}_k)^T$, where $\vec{\psi}_k$ is an eigenmode of $h_{\mathrm{HN}}$ for $L - 1$, with energy $E_k^{\mathrm{HN}}$. Then, from Eq. (D.4) it follows that

$$iV\nabla\vec{\Phi}_k = (E_k^{\mathrm{HN}} - 2i\Gamma_S)\vec{\Phi}_k\,, \tag{D.6}$$

and, after multiplying both sides by $\nabla$,

$$i\nabla V(\nabla\vec{\Phi}_k) = h(\nabla\vec{\Phi}_k) = (E_k^{\text{HN}} - 2i\Gamma_S)\nabla\vec{\Phi}_k. \tag{D.7}$$

This means that for each of the $L-1$ eigenfunctions $\vec{\psi}_k$ of $h_{\text{HN}}$ we obtain an eigenvector of $h$, with a modefunction $\nabla\vec{\Phi}_k$, and eigenenergy $E_k = E_k^{\text{HN}} - 2i\Gamma_S$. Moreover, since $\det(V) = 0$, there is one additional eigenstate $\psi_{\text{ss}}$ with energy $E_{\text{ss}} = 0$, which satisfies $V\psi_{\text{ss}} = h\psi_{\text{ss}} = 0$. It is straightfoward to check that this eigenstate is of the form given in Eq. (51). Putting these two sets of eigenstates together, we finally obtain the result of Eq. (50).

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
