# Peer review of "The bosonic skin effect: boundary condensation in asymmetric transport"

_SciPost Physics, doi:SciPost Phys. 16, 029 (2024)_

## Round 1 · Referee Report · Anonymous (Referee 1) · 2023-9-1

Strengths

1. A comprehensive study of incoherent transport of bosons through tilted lattice.
2. Mean field results supported by Monte Carlo and truncated Wigner.
3. Different transport phases identified with e.g. the phase with unusual, zig-zag occupation pattern.
4. Discovering the connection with non-Hermitian skin effect.

Weaknesses

Practically none. One would be happy to read about possible experimental schemes which could verify the findings in more detail.

Report

Journal criteria are met. The manuscript delivers important discovery that may influence experiments in the field. So the first criterium seems to be satisfied. All necessary acceptance criteria too. The manuscript is very carefully written with minor problems listed below.

Requested changes

1. p.6 just after Eq.(4). Either change "Were" to "Here" or replace dot with comma in (5) and "Were" to "where". In either case remove the indent probably created by an empty line after (5).
2. Similarly remove indent after (7).

  • validity: high
  • significance: high
  • originality: top
  • clarity: top
  • formatting: excellent
  • grammar: perfect

Author:  Louis Garbe  on 2023-10-04  [id 4027]

(in reply to Report 1 on 2023-09-01)

We thank the referee for taking the time to evaluate our work. We are grateful for their very positive assessment of our manuscript, in particular regarding its clarity and originality. We also thank them for their suggested changes, which we implemented in the updated version of our work.

---

## Round 1 · Referee Report · Anonymous (Referee 2) · 2023-9-7

Report

In this work, the Authors investigate a 1D bsonic lattice where, at each site, there is the possibility that a particle hops towards the left or the right side with different probabilities. Reservoirs are connected to the lattice to the left and right. Depending on the parameters of the system, the Authors investigate different regimes that showcase the interest of the setup and the richness of phenomena that can be observed.

It is a pleasure to read a manuscript that is so well written, interesting, clear and effective. While my field of expertise is not directly encompassing particle transport (and as such I cannot fully judge the relevance and/or importance of the results), I followed the derivation and truly enjoyed learning about asymmetric transport. I fully endorse this work for publication, and congratulate for its overall quality.

I highlight that by no means this review indicates a lack of time/effort on my side. I carefully read through the manuscript and derivations and am fully confident in my endorsement. Very minor suggestions are in the "Requested changes" section.

Requested changes

1- I would find aesthetically a bit more pleasant to collect references together when citing. E.g., [1-4] instead of [1,2,3,4].
2- When reading for the first time, I was a bit confused why the hopping dynamics is not expressed as a coherent dynamics, i.e., [\rho, H] in the master equation. This is very well addressed in the following sections and particularly in the appendices, but maybe the Authors could spend couple of sentences more motivating the form of Eq. (1) and (2).
3- After Eq. (7), there is a seemingly wrong indentation.
4- Please check the sentence after Eq. (14)... "is" is seemingly misplaced.
5- Perhaps use \Gamma_A/\Gamma_S instead of \Gamma_A/\Gamma_l in Fig. 2 and related discussion.
6- when introducing \tau_coh for the first time, after Eq. (30), it would be useful to refer to Eq. (30) later on, where it is formally defined.
7- In the second paragraph of Sec. 5.4, the first quotation mark in 'smooth' is reversed.

---

## Round 2 · Referee Report · Anonymous (Referee 2) · 2023-11-21

Report

Publish as soon as possible :)

---

## Round 2 · Referee Report · Anonymous (Referee 1) · 2023-11-21

Report

The criteria are met.

---

## Round 2 · Author Response

We would like to thank the editors for their consideration of our manuscript, and both referees for their time and efforts in evaluating our work. We are grateful for their very positive assessment of our manuscript, in particular regarding its clarity and originality.

We also thank them for their suggested changes, which helped us improve the readability of our paper. We implemented all of them in the resubmitted version (see list of changes below), except for Referee 2's suggestion of replacing \Gamma_A/\Gamma_l by \Gamma_A/\Gamma_S in Fig.2 and related discussions; we decided to keep our original convention because it gives many of the expressions (in particular Eq.26) a more transparent form.

With these changes, we hope our work is now suitable for publication in SciPost Physics.

---

## Round 2 · List of Changes

Paragraph added in Section 2.1 to clarify the use of an incoherent master equation; change of references formatting; reference to the definition of g(\tau) added in the caption of Fig.4; removal of unneeded indentations and minor typographic corrections throughout.

---

## Editorial Decision

published